# Preference Alignment with Flow Matching

**Minu Kim**[1][*]   **Yongsik Lee**[1][*]   **Sehyeok Kang**[1]   **Jihwan Oh**[1]
**Song Chong**[1][†]   **Se-Young Yun**[1][†]
[1]KAIST AI

{minu.kim, dldydtlr93, kangsehyeok0329, ericoh929,
songchong, yunseyoung}@kaist.ac.kr

## Abstract

We present Preference Flow Matching (PFM), a new framework for preference alignment that streamlines the integration of preferences into an arbitrary class of pre-trained models. Existing alignment methods require fine-tuning pre-trained models, which presents challenges such as scalability, inefficiency, and the need for model modifications, especially with black-box APIs like GPT-4. In contrast, PFM utilizes flow matching techniques to directly learn from preference data, thereby reducing the dependency on extensive fine-tuning of pre-trained models. By leveraging flow-based models, PFM transforms less preferred data into preferred outcomes, and effectively aligns model outputs with human preferences without relying on explicit or implicit reward function estimation, thus avoiding common issues like overfitting in reward models. We provide theoretical insights that support our method's alignment with standard preference alignment objectives. Experimental results indicate the practical effectiveness of our method, offering a new direction in aligning a pre-trained model to preference. Our code is available at https://github.com/jadehaus/preference-flow-matching.

## 1   Introduction

Preference-based reinforcement learning (PbRL) has emerged as a groundbreaking approach with significant contributions to performance improvement [Akrour et al., 2011, Wilson et al., 2012], particularly in the realm of artificial intelligence where understanding and incorporating human preferences are crucial. Unlike traditional reinforcement learning, which struggles due to the absence of explicit reward functions or the infeasibility of defining comprehensive environmental rewards, PbRL leverages a variety of feedback forms from humans to guide the learning process. This class of PbRL method is often referred to as reinforcement learning from human feedback (RLHF) [Ziegler et al., 2019, Levine et al., 2018, Ouyang et al., 2022].

Despite their effectiveness, these methods necessitate fine-tuning pre-trained models to align with user preferences, introducing several challenges such as scalability, accessibility, inefficiency, and the need for model modifications. For instance, with black-box APIs like GPT-4 [OpenAI et al., 2024], customization based on user preferences is constrained due to restricted access to the underlying model. Moreover, even if fine-tuning were feasible, the large model size results in inefficient training and high resource consumption. Aligning black-box models with user preferences remains an under-explored area in research, despite its critical importance and growing demand.

In this line of research, we propose Preference Flow Matching (PFM), which redefines the integration of human preferences by directly learning a preference flow from the less preferred data to the more preferred ones. This direct modeling of preference flows allows our system to better characterize

---

[*]Equal contribution.   [†]Corresponding authors.

38th Conference on Neural Information Processing Systems (NeurIPS 2024).

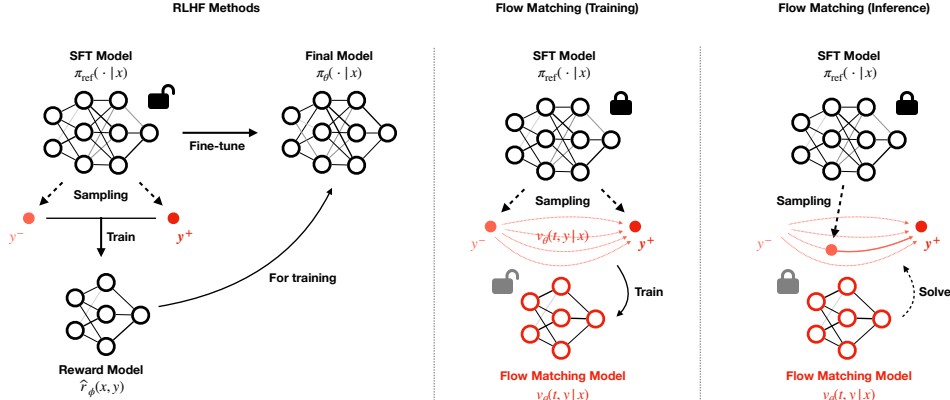

Figure 1: Illustration of our PFM framework. In the typical RLHF scenarios (left), we first sample preference data from the supervised fine-tuned (SFT) reference model. A reward model is learned from the collected dataset, either implicitly (as in DPO) or explicitly. The reward model is then used to fine-tune the reference policy to obtain the final model. Our method directly learns the preference flow from the collected preference data, where the flow is represented as a vector field $v_\theta$ (middle). For inference, we again sample a point from the reference policy, and improve the quality of alignment by using the trained flow matching model, without the need of fine-tuning the existing reference model (right).

and replicate the marginal distribution of the favored outcomes. We adopt a novel flow matching framework [Lipman et al., 2022], which is a simple, intuitive, yet relatively under-explored method for preference alignment. By simply adding a preference flow matching module to black-box models, PFM eliminates the need for fine-tuning the black-box model itself, providing a significant advantage.

Additionally, our method offers a highly robust approach for preference alignment, by circumventing the need for explicit or implicit reward function estimation. In typical RLHF scenarios, a model is initially trained to approximate a reward function based on human preferences. This reward model is then used to guide the policy learning process, aiming to align agent behaviors more closely with human preferences. However, this approach can introduce complexities and potential biases in translating human preferences into numerical rewards. In particular, learned reward models can often overfit the ground truth preference model, especially in the finite data regime [Azar et al., 2023]. Recent advancements such as Direct Preference Optimization (DPO) [Rafailov et al., 2024b] address the complexities of RLHF by eliminating the need for reward learning. However, these methods still inherently optimize for the reward model, and hence they are also susceptible to reward overfitting. In contrast, PFM directly learns preference flow, thereby removing the need for any reward model assumptions and resolving the challenges associated with reward model learning.

We prove both theoretically and empirically that our method is able to learn an object that is similar to the standard RLHF objectives, while being robust to the preference overfitting observed in traditional RLHF pipelines. We also demonstrate how we can further improve the quality of alignment via iterative flow matching, with theoretical guarantees. Experimentally, we find that while typical RLHF methods and DPO suffer from preference overfitting, our method can robustly align with preference and still achieve comparable performances.

## 2 Preliminaries

### 2.1 Reinforcement Learning from Human Feedback (RLHF)

Reinforcement learning from human feedback generally begins with obtaining a pre-trained reference policy $\pi_{\text{ref}}$ that can generate samples $y \sim \pi_{\text{ref}}(\cdot|x)$ given a context $x$. For example, a context $x$ could be a text prompt given by a user, and the sample $y$ could represent an appropriate text response generated by the reference policy $\pi_{\text{ref}}$. We then collect a dataset of $N$ preference pairs $\mathcal{D} = \{(x_i, y_i^+, y_i^-)\}_{i=1}^N$, where each $x_i$ denotes the context, and each $y_i^+, y_i^- \sim \pi_{\text{ref}}(\cdot|x_i)$ are generated responses to $x_i$ and marked as good or bad samples, respectively. Here, we assume that the preference $y_i^+ > y_i^-$ is generated from a ground-truth reward $r^* : X \times Y \to \mathbb{R}_{\geq 0}$, where $X$ and $Y$ are context space and response space, respectively. The goal of general RLHF is to recover an

optimal policy $\pi^*$ such that

$$\pi^* = \underset{\pi}{\arg\max} \; \mathbb{E}_x \left( \mathbb{E}_{y \sim \pi(\cdot|x)} \left( r^*(x,y) \right) - \beta \mathbb{D}_{\mathrm{KL}} \left( \pi(\cdot|x) \big\| \pi_{\mathrm{ref}}(\cdot|x) \right) \right). \tag{1}$$

RLHF pipelines generally require reward learning. One of the most popular choices of the reward model is the Bradley-Terry model [Bradley and Terry, 1952], which assumes that the preference $y^+ > y^-$ is generated from the probability $\mathbb{P}(y^+ > y^-|x) = \sigma(r^*(x,y^+) - r^*(x,y^-))$, where $\sigma$ is the logistic function. Under this model, the general RLHF framework learns the reward model $r_\phi \approx r^*$ by minimizing the negative log-likelihood:

$$\mathcal{L}_R(\phi; \mathcal{D}) := -\mathbb{E}_{(x,y^+,y^-) \sim \mathcal{D}} \left( \log \sigma \left( r_\phi(x, y^+) - r_\phi(x, y^-) \right) \right). \tag{2}$$

Once the reward model $r_\phi$ is trained, we then use it to optimize for (1) to obtain $\pi_\theta \approx \pi^*$ using standard reinforcement learning algorithms.

There is also a class of *reward-free* methods that eliminates the need of reward learning phase [Rafailov et al., 2024b, Azar et al., 2023]. Direct Policy Optimization (DPO) [Rafailov et al., 2024b] is a representative reward-free method that optimizes for (1) directly without learning a reward model. Although being a reward-free method, DPO implicitly optimizes for the reward function as in (2), by replacing $\hat{r}_\theta(x,y) = \beta \log(\pi_\theta(y|x)/\pi_{\mathrm{ref}}(y|x))$ as the implicit reward estimate.

## 2.2 Flow Matching

Flow matching is a class of generative model, where given a prior distribution $p_0$, we aim to model a target distribution $p_1$ from $p_0$. A key difference of the flow matching to the other generative models is that the prior $p_0$ can be an *arbitrary* distribution, (diffusion, for example, starts from a Gaussian prior $p_0$) and that the flow matching algorithm learns to modify the prior distribution $p_0$ to the target distribution $p_1$ with a neural network.

Throughout, we consider a pair of data distributions over $\mathbb{R}^d$ with densities $y^- \sim p_0$ and $y^+ \sim p_1$, possibly unknown (but able to sample). The flow matching considers the task of fitting a mapping $f : \mathbb{R}^d \to \mathbb{R}^d$ that transforms $p_0$ to $p_1$, that is, if $y^- \sim p_0$, then $f(y^-) \sim p_1$. Inspired as in the motivation for the diffusion models, one can define a smooth time-varying vector field $u : [0,1] \times \mathbb{R}^d \to \mathbb{R}^d$ that defines an ordinary differential equation (ODE),

$$dy = u(t, y)dt \tag{3}$$

where we use the notation $u(t, y)$ interchanably with $u_t(y)$. Denote the solution of the above ODE by $\phi(t, y)$ (or $\phi_t(y)$) with initial condition $\phi_0(y) = y$. In other words, $\phi_t(y)$ is the point $y$ transported along the vector field $u$ from time 0 to $t$. In order to obtain samples from the target distribution $p_1$, we simply compute $\phi_1(y)$ where $y \sim p_0$. The integration map $\phi_t$ induces a pushforward measure $p_t \triangleq [\phi_t]_\sharp(p_0)$, which is the density of points $y \sim p_0$ transported via $u$ from 0 to $t$.

To train the vector field $v_\theta$ with a neural network that mimics the vector field $u$ of our interest, we can solve for the conditional flow matching objective, as proposed by Lipman et al. [2022]:

$$\mathcal{L}_{\mathrm{CFM}}(\theta) \triangleq \mathbb{E}_{t \sim [0,1], z \sim q(\cdot), y \sim p_t(\cdot|z)} \left( \|v_\theta(t, y) - u_t(y|z)\|^2 \right) \tag{4}$$

where $q(z) = \pi(y^-, y^+)$ is some coupled distribution of samples $y^-, y^+$ and $u_t(y|z) = y^+ - y^-$ is a straight path from a source sample to a target sample. The conditional distribution $q(z)$ can be chosen to be an independent coupling of source and target distribution $q(z) = p_0(y^-)p_1(y^+)$ [Lipman et al., 2022], or the 2-Wasserstein optimal transport plan as proposed by Tong et al. [2023].

## 3 Preference Flow Matching

In this section, we describe how we can use flow matching to learn (human) preferences. In the first subsection, we illustrate our flow matching framework for learning preferences, and compare it with typical RLHF pipelines. Then in Section 3.2, we demonstrate our method in a simple 2-dimensional toy experiment. Finally in Section 3.3, we provide an extension of our framework that can iteratively improve the performance, with theoretical guarantees.

### 3.1 Flow Matching for Preference Learning

Instead of trying to optimize for the unknown reward $r^*$ or the preference probability model $\mathbb{P}(y^+ > y^- | x)$, we simply learn a *flow* from the marginal distribution of *less preferred data* $p_0(y^- | x)$ to the marginal distribution of *more preferred data* $p_1(y^+ | x)$ by leveraging what is explicitly characterized in the collected preference data:

$$p_0(y^- | x) \propto \pi_{\text{ref}}(y^- | x) \int \pi_{\text{ref}}(y | x) \mathbb{P}(y > y^- | x) dy \tag{5}$$

$$p_1(y^+ | x) \propto \pi_{\text{ref}}(y^+ | x) \int \pi_{\text{ref}}(y | x) \mathbb{P}(y^+ > y | x) dy \tag{6}$$

$$= \pi_{\text{ref}}(y^+ | x) \, \mathbb{E}_{y \sim \pi_{\text{ref}}(\cdot | x)} \left( \mathbb{P}(y^+ > y | x) \right) \tag{7}$$

In other words, we view that our collected data $\mathcal{D}$ is in fact generated from each of the marginal distributions $y^- \sim p_0(\cdot | x)$ and $y^+ \sim p_1(\cdot | x)$ obtained from $\mathbb{P}(y^+ > y^- | x)$, respectively. Hence, following the conventions in the literature, [Tong et al., 2023] we define the flow matching objective for preference dataset $\mathcal{D}$ as follows:

$$\mathcal{L}(\theta) = \mathbb{E}_{t \sim [0,1], z \sim \mathcal{D}, y \sim p_t(\cdot | z)} \left( \| v_\theta(t, y | x) - u_t(y | z) \|^2 \right) \tag{8}$$

where we define the condition $z = (x, y^+, y^-)$, conditional flow $u_t(y | z) = y^+ - y^-$, and the probability path $p_t(y | z) = \mathcal{N}(y | ty^+ + (1 - t)y^-, \sigma^2)$. Once we obtain the vector field $v_\theta$, we can improve upon the generated negative samples $y^- \sim p_0(\cdot | x)$ by solving (3) using an off-the-shelf numerical ODE solver [Runge, 1895, Kutta, 1901] to obtain samples $f(y^-) \sim p_1$. Specifically, we start from a sample $y^-$ with $t = 0$, and "flow" along the ODE trajectory using $v_\theta$ until $t = 1$, to arrive at the target $y^+$. Detailed algorithm can be found in Algorithm 1. Notably, generating improved samples can be done without the need of fine-tuning the existing model, since we learn a separate vector field that transports negative samples from $p_0$ to $p_1$. Furthermore, we did not require any assumption for the probability model $\mathbb{P}(y^+ > y^- | x)$, so our method can extend to general scenarios that do not adopt the Bradley-Terry model. Our method is outlined in Figure 1.

A careful reader might notice that for inference, we require negative samples $y^-$ from the marginal distribution $p_0$, to obtain aligned samples $y^+$. However, this $p_0$ is inaccessible during inference step, as we must first acquire preference label $y^+ > y^-$ for samples generated from $\pi_{\text{ref}}$. Instead, we simply start from $y \sim \pi_{\text{ref}}$ as the starting point, and apply flow matching to obtain $f(y) \approx y^+ \sim p_1$. We emphasize that PFM can still robustly generate positive samples, if we assume non-deterministic preferences *i.e.*, $\text{supp}(p_1) \supseteq \text{supp}(p_0)$. We also empirically find that using $\pi_{\text{ref}}$ instead of $p_0$ as the source distribution can produce comparable results in practical scenarios. Further details can be found in Appendix B.

### 3.2 Illustrative Example: Preference Generated from 8-Gaussians Density

Here, we demonstrate how our method learns to improve generated samples to better align with the preference, in a simple 2-dimensional toy experiment. We consider a ground truth reward function generated from an 8-Gaussians density as illustrated in Figure 2a. We then pre-train a Gaussian mixture model to obtain samples as in Figure 2c. The pairwise preference labels are then generated using the ground truth 8-Gaussians reward function, as done in many existing preference-based reinforcement learning (PbRL) settings [Christiano et al., 2017, Ibarz et al., 2018, Shin et al., 2023].

Once preference data are collected, we first learn a reward model $\widehat{r}_\phi$ via (2). As can be seen in Figure 2b, the learned reward model overfits in the unseen region, which causes the RLHF method to diverge (Figure 2e). DPO also fails to learn the correct preference, as can be seen in Figure 2f. We note here that DPO is also subjective to the reward overfitting since DPO also implicitly learns to optimize for the reward using the Bradley-Terry model (2) [Xu et al., 2024, Azar et al., 2023].

However, PFM is free of such reward overfitting issues, as we do not optimize for the reward function using the Bradley-Terry model. Unlike other RLHF methods, our model robustly learns to align with the preference from the provided dataset (Figure 2g). Notably, our method does not try to overfit beyond the unseen region, since the learned target distribution from the flow matching model tries to mimic the distribution $p_1(y^+)$ of collected preferred samples. (Compare Figure 2d and Figure 2g.)

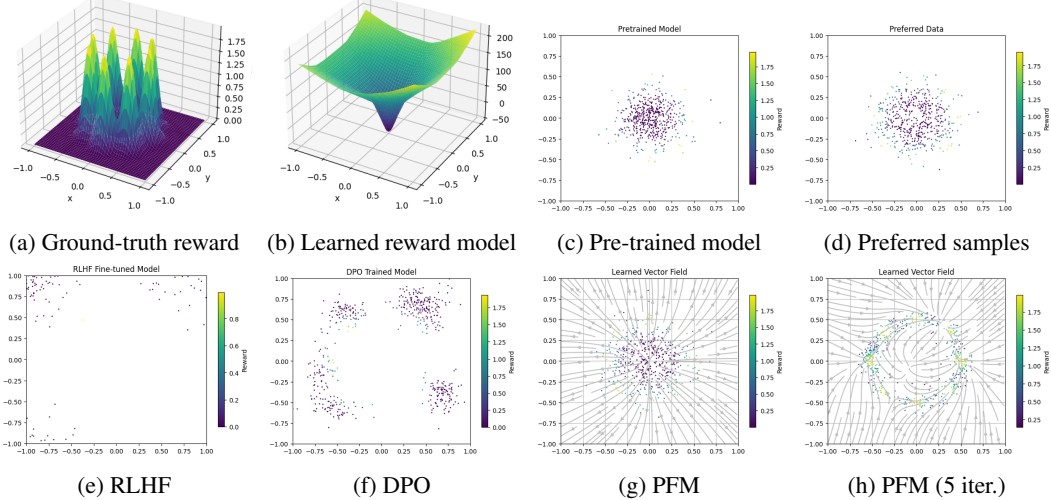

| (a) Ground-truth reward | (b) Learned reward model | (c) Pre-trained model | (d) Preferred samples |
|---|---|---|---|
| (e) RLHF | (f) DPO | (g) PFM | (h) PFM (5 iter.) |

Figure 2: Comparison of RLHF, DPO, and PFM on a 2-dimensional toy experiment. We generate preference labels from a ground truth reward in (a) and a pre-trained Gaussian reference policy (c). Both the RLHF (e) and DPO (f) methods struggle to align with the preferences, due to the overfitted reward model (b), even with the presence of KL regularizer ($\beta = 1$). PFM is able to mimic the distribution of the positively-labeled samples (d), and therefore achieves the highest performance (g). Repeating PFM iteratively to the marginal samples can further improve the alignment with the preference (h).

## 3.3 Improving Alignment with Iterative Flow Matching

As done in iterative variants of DPO [Xiong et al., 2023, Yuan et al., 2024], we can also further improve the quality of alignment with iterative flow matching. Specifically, upon obtaining a marginal distribution $p_1$ by applying flow matching, we again collect a new preference data $y^-, y^+$ from the obtained marginal distribution $p_1$ in (6). We repeat this process iteratively, by replacing the source distribution (which is $\pi_{\text{ref}}$ in the first step) with the marginal distribution $p_1$ obtained in the latest iteration. This iterative process can be summarized as follows.

$$p_0^{(n)}(y^-|x) \propto p_1^{(n-1)}(y^-|x) \int p_1^{(n-1)}(y|x)\mathbb{P}(y > y^-|x)dy \tag{9}$$

$$p_1^{(n)}(y^+|x) \propto p_1^{(n-1)}(y^+|x) \int p_1^{(n-1)}(y|x)\mathbb{P}(y^+ > y|x)dy, \quad p_1^{(0)} = \pi_{\text{ref}}, \tag{10}$$

where we denote $p_0^{(n)}$ and $p_1^{(n)}$ to be the source and target distribution of the flow matching model at the $n$-th iteration, respectively. By repeatedly marginalizing the distribution with respect to the preference $\mathbb{P}(y^+ > y^-|x)$, we can effectively "narrow" the sampling distribution towards the outputs with higher preference probability. See Figure 2h for the results of the iterative method in our toy experiment. Note that even during this iterative approach, we leave the parameters of the pre-trained model $\pi_{\text{ref}}$ untouched, and only require sampling from this model throughout the whole process. Later in Section 4, we formally prove that the iterative method allows us to obtain a distribution class that maximizes the ground truth expected preference, and hence yields an optimal policy $\pi^*$ in (1) with $\beta = 0$. See Theorem 4.2.

## 4 Theoretical Analysis of Preference Flow

In this section, we theoretically analyze why PFM framework can effectively learn to align with the preference. Interestingly, learning to generate samples from the marginal distribution $p_1$ in (6) optimizes an objective that is similar to the goal of general RLHF in (1). Following the formulation provided by Azar et al. [2023], one can observe that the objective (1) is equivalent to the below form:

$$\pi^* = \underset{\pi}{\operatorname{argmax}} \, \mathbb{E}_{y \sim \pi} \left( \mathbb{E}_{y' \sim \pi_{\text{ref}}} \left( \sigma^{-1}(\mathbb{P}(y > y')) \right) \right) - \beta \mathbb{D}_{\text{KL}} \left( \pi \middle\| \pi_{\text{ref}} \right) \tag{11}$$

where $\sigma^{-1}(\xi) = \log(\xi/(1-\xi))$ is the logit function, and we drop the conditional dependence of $x$ for simplicity. Note DPO also optimizes for the same objective as in (11).

Let us take a step back, and characterize the failure modes of the RLHF and DPO frameworks, by figuring out when these methods overfit the reward. Consider a case where the preferences are *deterministic*, *i.e.*, $\mathbb{P}(y > y') = 1$, so that $y$ is always preferred to $y'$. If we plug it into (11), we see that $\sigma^{-1}(\mathbb{P}(y > y')) \to +\infty$. Therefore, the solution $\pi^*$ of (11) ends up overfitting for the preference likelihood, resulting in a weak or even null KL regularization, regardless of the size of $\beta$.

Although in the case where the preference is not deterministic, this phenomenon can still be pronounced in the finite data regime [Azar et al., 2023]. Even if the true preference is strictly less than 1, we may have access to only a few data samples to estimate $\mathbb{P}(y > y') \approx 1$. This means that overfitting can be a critical issue in general, especially if the action space $Y$ or the context space $X$ is extremely large, as in the case of aligning large language models to human preferences.

In contrast, the PFM framework learns to generate a marginal distribution $p_1$. One can show that this marginal is a solution for the optimization problem that is similar to the objective (11).

**Theorem 4.1** (**Characterization of the Marginal**). *Let $p_1$ denote the marginal distribution of the positively-labeled samples $y^+$. Then the marginal distribution $p_1$ obtained from the preference model $\mathbb{P}(y > y'|x)$ is an optimizer of the optimization problem*

$$p_1 = \underset{\pi}{\arg\max} \ \mathbb{E}_{y \sim \pi}\bigg( \log \mathbb{E}_{y' \sim \pi_{\text{ref}}}\Big(\mathbb{P}(y > y')\Big)\bigg) - \mathbb{D}_{\text{KL}}\Big(\pi \Big\| \pi_{\text{ref}}\Big). \tag{12}$$

We defer the proof to the Appendix C. Similar to the RLHF and DPO objective (11), the solution $p_1$ of (12) drives the original distribution $\pi_{\text{ref}}$ towards the points where the preference probability $\mathbb{P}(y > y')$ is increasing. However, unlike the RLHF or DPO objectives, the objective (12) is bounded even in the deterministic case $\mathbb{P}(y > y') = 1$, making it robust to reward overfitting.

Interestingly, maximizing the objective (12) is equivalent to minimizing the KL distance between the policy $\pi$ and the normalized preference score with a cross-entropy constraint:

$$p_1 = \underset{\pi}{\arg\min} \ \mathbb{D}_{\text{KL}}(\pi \| \widetilde{\mathbb{P}}) - \mathbb{E}_{y \sim \pi}(\log \pi_{\text{ref}}(y)) \quad \text{where} \quad \widetilde{\mathbb{P}}(y) \propto \mathbb{E}_{y' \sim \pi_{\text{ref}}}(\mathbb{P}(y > y')). \tag{13}$$

Hence, the objective pushes the policy $\pi$ to match the preference $\widetilde{\mathbb{P}}$, while encouraging $\pi$ to align with the high-probability samples in $\pi_{\text{ref}}$. Since the constraint restricts the policy to the high-probability regions of $\pi_{\text{ref}}$ where the preference labels are collected from, our method is less prone to reward overfitting in the out-of-distribution samples due to the distribution shift in $\pi$ from $\pi_{\text{ref}}$ [Gao et al., 2023, Rafailov et al., 2024a]. Though we find this information-theoretic formulation interesting, we leave further analysis to future work.

Despite its robustness, one may notice that the objective (12) is less flexible compared to the original objective, due to the fixed regularization constant with $\beta = 1$. Below, we show that if we apply the iterative algorithm provided in Section 3.3, one can further reduce the KL regularization strength and obtain an optimal policy $\pi^*$ in (11) with $\beta \to 0$.

**Theorem 4.2** (**Convergence of Iterative Method**). *Assume $\pi_{\text{ref}} \in L^2$ and $\mathbb{P}(y > y') \in L^2$. Consider an iterative update of the marginal distribution $p_1$ in (10). Then, the iteration converges to the uniform distribution of points $y$ where the value $\mathbb{E}_{y^- \sim \pi_{\text{ref}}}(\mathbb{P}(y > y^-))$ is the largest, i.e.,*

$$p_1^{(\infty)} \to U\left(\left\{y : y \in \underset{y}{\arg\max} \ \mathbb{E}_{y^- \sim \pi_{\text{ref}}}\big(\mathbb{P}(y > y^-)\big)\right\}\right), \tag{14}$$

*where $U$ stands for uniform distribution, and we drop the conditional dependence of $x$ for simplicity.*

We defer the proof to the Appendix C. Intuitively, the proof follows from the fact that the marginalization iteratively "narrows" down the distribution towards the outputs with higher preference. We note here that the $L^2$ assumptions are generally valid in practical domains. See Appendix C.

## 5 Experimental Results

In this section, we conduct experiments to address the following questions: *Q1. Can PFM align generated samples from the black-box model with preference and achieve comparable results in practical tasks? Q2. Is PFM more beneficial than methods optimizing for explicit/implicit reward model?* and *Q3. Is PFM more beneficial than naïve add-on methods, e.g., separately training generative models to imitate preferred samples?* To answer these questions, we validate our method in three domains: Conditional text and image generation, and offline reinforcement learning tasks.

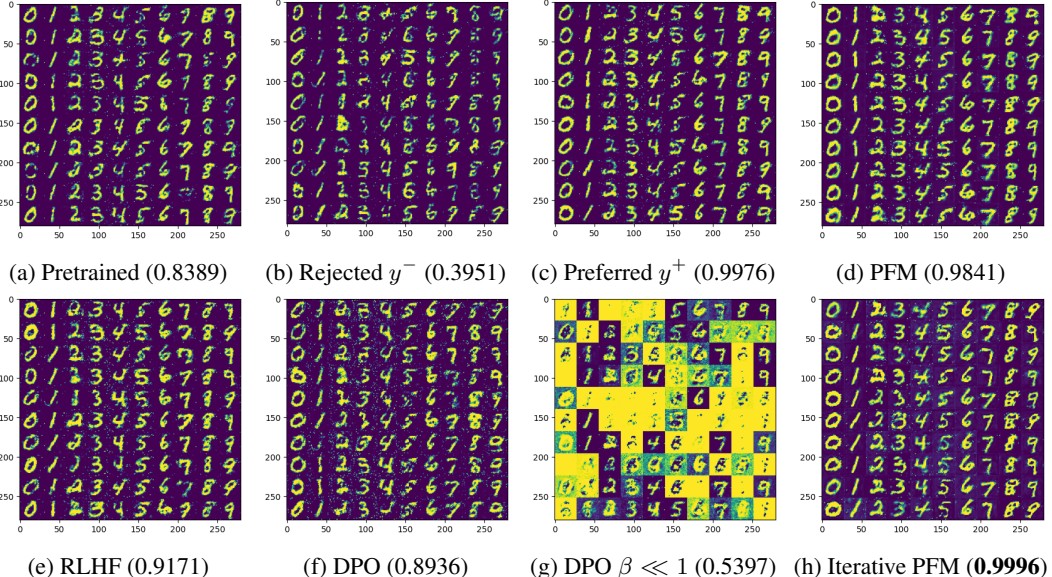

| (a) Pretrained (0.8389) | (b) Rejected $y^-$ (0.3951) | (c) Preferred $y^+$ (0.9976) | (d) PFM (0.9841) |

| (e) RLHF (0.9171) | (f) DPO (0.8936) | (g) DPO $\beta \ll 1$ (0.5397) | (h) Iterative PFM (**0.9996**) |

Figure 3: Comparison of RLHF, DPO, and PFM on a conditional MNIST image generation task. Numbers represent the preference score. PFM (d) demonstrates superior sample quality and preference alignment compared to RLHF (e) and DPO (f), where DPO collapses with a small size of $\beta$ (g). The iterative PFM with only two iterations (h) results in almost perfectly aligning with the preferences.

## 5.1 Conditional Image Generation

We first evaluate PFM on a conditional image generation task using the MNIST dataset [LeCun et al., 1998]. Specifically, we utilize a pre-trained DCGAN [Radford et al., 2015] generator as $\pi_{\text{ref}}$ and collect sample pairs from $\pi_{\text{ref}}(\cdot|x)$ conditioned on the digit labels $x \in \{0, \cdots, 9\}$. To construct preference datasets, we assign preferences to sample pairs according to the softmax probabilities of the labels from a LeNet [LeCun et al., 1998]. Then, we learn a PFM flow $v_\theta$ that transports $y^-$ to $y^+$ given a condition $x$. More experimental details are provided in the Appendix D.

Figure 3a illustrates the generated samples from $\pi_{\text{ref}}$, and the rejected and preferred images are depicted in Figure 3b and Figure 3c, respectively, where the values in parenthesis are the measured preference score. As shown in Figure 3d, PFM achieves higher preference alignment and better sample quality than RLHF (Figure 3e) and DPO (Figure 3f) without fine-tuning $\pi_{\text{ref}}$. Moreover, PFM achieves nearly perfect alignment with the preferences after only two iterations (Figure 3h), demonstrating the effectiveness of iterative PFM.

## 5.2 Conditional Text Generation

Next, we adopt a controlled (positive) sentiment review generation task. As done in Rafailov et al. [2024b], to perform a controlled evaluation, we adopt the pre-trained sentiment classifier as the preference annotator. We train a preference flow on randomly selected pairs of movie reviews $y^+, y^-$ from the IMDB dataset [Maas et al., 2011]. For our PFM framework to be applied to variable-length inputs, we employ a T5-based autoencoder to work with fixed-sized embeddings. We adopt GPT-2 SFT model on the IMDB dataset as a reference model $\pi_{\text{ref}}$. We also compare our method with a RLHF (PPO) fine-tuned policy $\pi_{\text{PPO}}$. See Appendix D for detailed experimental settings.

Below, we report the average preference score (from the classifier annotator) of 100 randomly generated review samples for each method. As shown in Table 1, PFM is able to improve the preference score of any baseline model to which it is attached. We emphasize here that our method requires relatively smaller training cost compared to the standard RLHF frameworks, even in the iterative settings. See Table 6 in Appendix D for the number of parameters that require training for each framework in tackling this task. PFM requires training a much smaller number of parameters (around 1.2%) while still achieving better performance. We also note here that instead of iteratively training the PFM module as described in Section 3.3, simply applying the same learned preference flow iteratively to the improved samples achieves the best performance. (See Appendix D.)

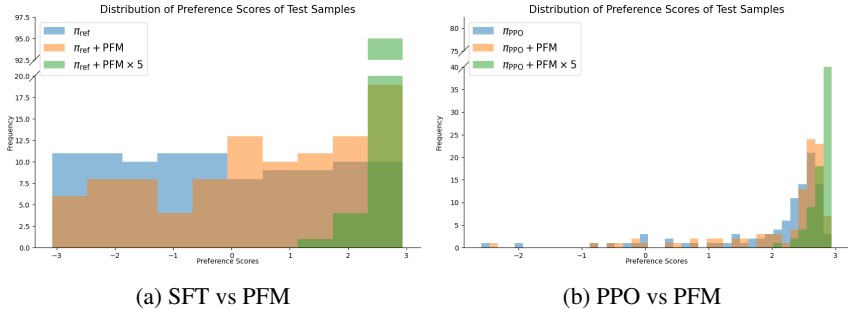

|                    |                    |
|:------------------:|:------------------:|
| (a) SFT vs PFM     | (b) PPO vs PFM     |

Figure 4: Distribution of preference scores for each method. Left visualizes the distribution of scores for the pre-trained reference policy and PFM-attached policy. Without fine-tuning the reference policy, PFM can obtain substantially better results by only adding a small flow-matching module. Right visualizes the preference score distribution of the RLHF (PPO) fine-tuned policy, and the PFM added policy to the PPO fine-tuned policy. Note that PFM is trained with the original dataset, not by the dataset generated from the PPO fine-tuned policy.

Table 1: Average preference scores of 100 test instances.

| $\pi_{\mathrm{ref}}$ | $\pi_{\mathrm{ref}} + \mathrm{PFM}$ | $\pi_{\mathrm{ref}} + \mathrm{PFM} \times 5$ |
|---|---|---|
| -0.3607 | 0.6399 | **2.7469** |

| $\pi_{\mathrm{PPO}}$ | $\pi_{\mathrm{PPO}} + \mathrm{PFM}$ | $\pi_{\mathrm{PPO}} + \mathrm{PFM} \times 5$ |
|---|---|---|
| 2.0156 | 2.178 | **2.7894** |

Table 2: GPT-4 win rate over 100 test samples.

| Method | vs. $\pi_{\mathrm{ref}}$ | vs. $\pi_{\mathrm{PPO}}$ |
|---|---|---|
| $\pi_{\mathrm{ref}} + \mathrm{PFM}$ | **100%** | 2% |
| $\pi_{\mathrm{ref}} + \mathrm{PFM} \times 5$ | – | 85% |
| $\pi_{\mathrm{PPO}}$ | **100%** | – |
| $\pi_{\mathrm{PPO}} + \mathrm{PFM}$ | – | 99% |
| $\pi_{\mathrm{PPO}} + \mathrm{PFM} \times 5$ | – | **100%** |

Interestingly, we observe that the distribution of the scores tends to shift more toward that of the preferred samples with an increasing number of PFM iterations. (See Figure 4.) This result aligns with our theoretical insights: the PFM framework learns to shift the source distribution (i.e., the distribution of the reference policy) toward the marginal distribution of the more preferred samples.

We also compute the win rate with GPT-4 evaluation. The results are summarized in Table 2. Both PFM and PPO fine-tuned policy excel the reference policy with win rates 100%. (First column of Table 2.) Furthermore, we observe that the iterative PFM with 5 iterations on the reference model outperforms the PPO fine-tuned policy. If PFM is added on top of the PPO fine-tuned policy, we observe near 100% win rates for both PPO + PFM and PPO + Iterative PFM.

## 5.3 Offline Reinforcement Learning

Finally, we employ the D4RL [Fu et al., 2020] benchmark to assess the performance of PFM in reinforcement learning tasks. Following the prior works on the PbRL literature, we adopt trajectory-based preference alignment [Hejna et al., 2023, Kim et al., 2023]. We first randomly choose an starting state $s_0 \sim S$, and sample two trajectories $\tau^+$ and $\tau^-$ with fixed length $\ell \geq 2$ from $\pi_{\mathrm{ref}}$:

$$\tau^+ := (a_0, a_1, \cdots, a_\ell) \sim \pi_{\mathrm{ref}}(\cdot | s_0) \tag{15}$$

$$\tau^- := (a'_0, a'_1, \cdots, a'_\ell) \sim \pi_{\mathrm{ref}}(\cdot | s_0). \tag{16}$$

Then, we obtain the preference $\tau^+ > \tau^-$ given the starting state $s_0$ using a scripted teacher approach that has also been widely adopted in the PbRL settings [Lee et al., 2021, Kim et al., 2023], which prioritizes trajectories with higher rewards based on the ground truth reward. For inference at a given state $s_t$, we again sample an action trajectory $\tau = (a_t, \cdots, a_{t+\ell})$ from $\pi_{\mathrm{ref}}(\cdot | s_t)$, and apply flow matching to obtain a better action sequence.

The baseline methods for comparing the performance of PFM include behavior cloning (BC), which we adopt as our pre-trained reference model $\pi_{\mathrm{ref}}$, and a DPO fine-tuned model from the BC model. Additionally, we train a separate behavior cloning model to the collected preferred samples $y^+ \sim p_1$, aiming to replicate the marginal distribution of the "good" trajectories. Further experimental details are deferred to Appendix D.

Table 3 presents the outcomes from evaluations conducted on 12 offline datasets. Our findings indicate that PFM consistently demonstrates comparable or even superior performance with lower

Table 3: Normalized results on MuJoCo datasets. Mean and standard deviation from 5 seeds are reported.

| Dataset | BC | DPO Fine-tuned | PFM (Ours) | Marginal BC |
|---------|-----|----------------|------------|-------------|
| ant-random-v2 | $31.59 \pm 0.05$ | $31.52 \pm 0.08$ | $\mathbf{31.62} \pm \mathbf{0.13}$ | $25.01 \pm 4.64$ |
| ant-medium-v2 | $90.16 \pm 21.48$ | $95.04 \pm 13.93$ | $96.73 \pm 2.47$ | $\mathbf{99.67} \pm \mathbf{1.57}$ |
| ant-expert-v2 | $125.83 \pm 24.07$ | $\mathbf{134.96} \pm \mathbf{3.76}$ | $132.20 \pm 2.69$ | $99.29 \pm 34.74$ |
| hopper-random-v2 | $3.17 \pm 0.25$ | $3.23 \pm 0.25$ | $\mathbf{7.69} \pm \mathbf{0.08}$ | $5.48 \pm 4.46$ |
| hopper-medium-v2 | $52.83 \pm 5.03$ | $53.47 \pm 3.92$ | $\mathbf{58.76} \pm \mathbf{2.62}$ | $40.44 \pm 1.69$ |
| hopper-expert-v2 | $111.27 \pm 0.48$ | $111.51 \pm 0.92$ | $\mathbf{111.70} \pm \mathbf{0.77}$ | $32.39 \pm 0.1$ |
| halfcheetah-random-v2 | $2.25 \pm 0.01$ | $2.26 \pm 0.01$ | $\mathbf{2.26} \pm \mathbf{0.0}$ | $2.21 \pm 0.02$ |
| halfcheetah-medium-v2 | $40.97 \pm 0.89$ | $41.94 \pm 0.68$ | $\mathbf{43.49} \pm \mathbf{0.88}$ | $38.79 \pm 1.27$ |
| halfcheetah-expert-v2 | $91.02 \pm 1.24$ | $\mathbf{92.15} \pm \mathbf{0.76}$ | $90.05 \pm 0.83$ | $4.77 \pm 2.5$ |
| walker2d-random-v2 | $1.47 \pm 0.1$ | $1.38 \pm 0.08$ | $1.77 \pm 0.13$ | $\mathbf{2.45} \pm \mathbf{0.38}$ |
| walker2d-medium-v2 | $60.35 \pm 18.16$ | $\mathbf{74.05} \pm \mathbf{12.05}$ | $72.59 \pm 15.8$ | $65.29 \pm 12.58$ |
| walker2d-expert-v2 | $\mathbf{108.62} \pm \mathbf{0.39}$ | $108.38 \pm 0.28$ | $108.36 \pm 0.21$ | $15.8 \pm 0.54$ |
| Random Average | 9.62 | 9.60 | **10.84** | 8.79 |
| Medium Average | 61.08 | 66.13 | **67.89** | 61.05 |
| Expert Average | 109.19 | **111.75** | 110.58 | 38.06 |
| D4RL Average | 59.97 | 62.49 | **63.10** | 35.97 |

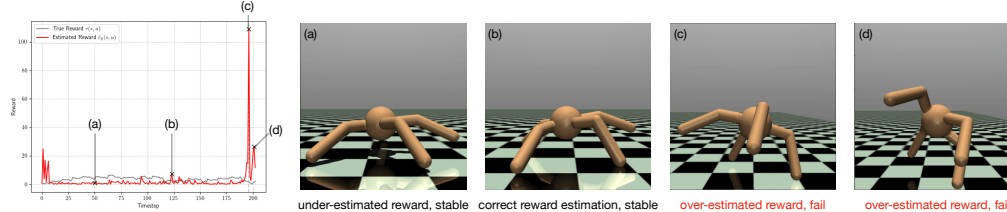

Figure 5: Analysis of a sample episode of a DPO fine-tuned model on the MuJoCo ant environment. DPO fine-tuned model often overestimates the reward due to reward overfitting (*e.g.*, $t = 196$). This can cause the policy to choose problematic actions. Here, the implicit reward estimation is $\hat{r}_\theta(s, a) = \beta \log(\pi_\theta(a|s)/\pi_{\text{ref}}(a|s))$.

variance across all baseline methods. Notably, our method excels particularly in datasets generated from suboptimal behavioral policies, achieving better performance. Furthermore, PFM manages to match the performance on expert datasets even in the absence of a reward model, underscoring its robustness and effectiveness. This demonstrates that PFM can effectively align black-box models with preference through flow matching, without the need to fine-tune the pre-trained model.

## 5.4 Is Learning a Flow Truly Beneficial?

In this remaining section, we focus on answering the remaining questions, *Q2* and *Q3*. We first investigate why PFM can be advantageous over previous methods with explicit/implicit reward modeling. As can be seen in Figure 5, DPO, like typical RLHF approaches, is also prone to reward overfitting, and may cause the agent to fail. This is because if the preference estimate is close to 0 or 1, these methods may end up overoptimizing to the exploding reward model [Ziegler et al., 2019, Gao et al., 2023]. PFM, on the other hand, is inherently robust to such over-estimation, as we adopt a completely different optimization framework that does not require a reward proxy. (See Theorem 4.1.)

On the other hand, we observe less performance gain in the expert datasets. This is a possible failure mode of PFM, where the generated samples are already near-optimal. In such regime, an arbitrary source $y \sim \pi_{\text{ref}}$ has a near zero probability of being sampled from the true marginal $p_0$, suggesting that PFM with prior as $\pi_{\text{ref}}$ might suffer from a shifted source distribution. We verify this experimentally on walker2d, where PFM struggles the most. By adopting a true marginal $p_0$ as the source, PFM with prior $p_0$ can achieve the highest performance among all baselines. This performance is evident even on the expert dataset, matching our theoretical analysis. See Appendix B.

Next, we compare PFM to an alternative approach that simply tries to approximate the marginal distribution directly from the positive samples. Intuitively, training a generative model from scratch that replicates the marginal $p_1$ is as computationally costly as training the original reference model. Experimentally, we observe that PFM achieves better performance than training a behavior cloning model (Marginal BC) to replicate the distribution of the preferred samples (Table 3). However, it is also worth mentioning that the Marginal BC model does occasionally yield the best results, suggesting the potential of using a marginal distribution for preference alignment.

## 6   Related Works

Contrastive Preference Learning (CPL) [Hejna et al., 2023] is a class of reward-free methods that utilizes contrastive learning techniques to align model outputs with the preferences observed in the dataset. By leveraging contrastive loss, CPL encourages the model to distinguish between preferred and less preferred outcomes effectively. Flow-to-Better (FTB) [Zhang et al., 2023] innovatively uses a diffusion model to transition from less preferred data to more preferred data, similar to the flow-based approach in our work. However, FTB mainly focuses on data augmentation, where they used the trained diffusion model to generate more data samples for behavior cloning. Despite their strengths, both works rely on the Bradley-Terry model to implicitly learn the reward function.

Identity Preference Optimization (IPO) [Azar et al., 2023] builds upon the foundation laid by DPO, extending the framework to accommodate a broader range of preference models beyond the Bradley-Terry paradigm. In particular, they focus on finding an objective that is bounded even in a deterministic preference regime, by replacing the function $\sigma^{-1}$ in (11) with an identity function $I(x) = x$. This effectively mitigates the reward overfitting problem observed in DPO and standard RLHF methods.

Our method distinguishes itself from these approaches by not requiring the Bradley-Terry assumption nor the fine-tuning of pre-trained models. This eliminates the risk of reward overfitting associated with the Bradley-Terry model and reduces the computational cost significantly. By avoiding the need for fine-tuning, our method offers a more efficient and scalable solution for integrating human preferences into reinforcement learning systems. This makes our approach particularly suitable for scenarios where computational resources are limited or where quick adaptation to human feedback is essential. The comparison of these related works is summarized in Table 4.

Table 4: Comparison of our method to other works.

|                                   | RLHF | DPO | IPO | CPL | FTB | PFM (Ours) |
|-----------------------------------|------|-----|-----|-----|-----|------------|
| Reward Model Free                 | ✗    | ✓   | ✓   | ✓   | ✓   | ✓          |
| No Reward Assumptions (e.g. BT)   | ✗    | ✗   | ✓   | ✗   | ✗   | ✓          |
| Applicable to Black-Box Models    | ✗    | ✗   | ✗   | ✗   | ✓   | ✓          |

## 7   Conclusion and Limitations

In conclusion, this research introduces Preference Flow Matching (PFM), a novel add-on approach that offers a practical, efficient, and scalable solution for integrating human preferences. This research highlights the potential of flow matching as a powerful tool for preference alignment and opens new avenues for further exploration and development in the field of RLHF. The ability to align black-box models with human preferences without extensive model modifications marks a critical step forward, with broad implications for the deployment and usability of AI systems in real-world applications.

Our theoretical and empirical analyses demonstrate that PFM achieves alignment performance comparable to standard RLHF methods while being more resilient to preference overfitting. The iterative flow matching technique further enhances alignment quality, by continually refining the preference alignment without modifying the underlying pre-trained model parameters.

Despite these promising results, the current design of the PFM framework entails several challenges in the natural language processing (NLP) domain. The PFM framework, as currently designed, relies on the autoencoder to work with fixed-sized embeddings to handle variable-length texts. To scale our method to more complex NLP tasks, future research should explore ways to adapt the PFM framework to long-form texts.

## Acknowledgements

This work was supported by Institute for Information & communications Technology Promotion(IITP) grant funded by the Korea government(MSIT)(No.RS-2019-II190075 Artificial Intelligence Graduate School Program(KAIST)), the Institute of Information & communications Technology Planning & Evaluation (IITP) grant funded by the Korea government (MSIT) (No. 2022-0-00871, Development of AI Autonomy and Knowledge Enhancement for AI Agent Collaboration), the National Research Foundation of Korea(NRF) grant funded by the Korea government(MSIT) (No. RS-2019-NR040050 Stochastic Analysis and Application Research Center (SAARC)), Institute of Information & communications Technology Planning & Evaluation (IITP) under Open RAN Education and Training Program(IITP-2024-RS-2024-00429088) grant funded by the Korea government(MSIT), and conducted by Center for Applied Research in Artificial Intelligence (CARAI) grant funded by DAPA and ADD (UD230017TD).

We thank Taehyeon Kim at KAIST for pointing out the strengths of our work and providing motivations of this work. We would also like to appreciate Sihyeon Kim and Yongjin Yang at KAIST for providing experimental details and suggesting relevant settings for PbRL tasks. Finally, we thank Junghyun Lee at KAIST for revising the details of the proofs of theorems.

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

# A  Flow Matching Algorithm for Preference Alignment

Below, we outline our method in Algorithm 1. Although the vector field $v_\theta$ is trained to transport the source distribution of less preferred samples $p_0$ to the target distribution, we use $\pi_{\mathrm{ref}}$ for inference instead of $p_0$ due to inaccessibility to the preference model.

Note that with the non-deterministic preference assumption, PFM is still theoretically guaranteed to obtain $\phi_1(y) \sim p_1$, even if the sample $y$ is obtained from $\pi_{\mathrm{ref}}$ instead of $p_0$. In particular, we assume $\mathrm{supp}(p_1) \supseteq \mathrm{supp}(p_0)$, *i.e.*, for any sample $y^+$ obtained from the target distribution $p_1$, the probability of $y \sim p_0$ is non-zero. Then, for any sample $y \sim \pi_{\mathrm{ref}}$ sampled from the reference model, we can guarantee that this sample has non-zero probability of being sampled as *less preferred*, indicating that there is a learned flow from this point to a better sample, with non-zero probability. In the next section, we provide empirical evidence that further justifies the use of $\pi_{\mathrm{ref}}$ instead of $p_0$ for inference.

---

**Algorithm 1:** PFM: Preference Flow Matching

1   *# Training*
2   **repeat**
3     *Sample $z = (x, y^+.y^-) \sim \mathcal{D}$*
4     $u_t(y|z) \leftarrow y^+ - y^-$
5     $p_t(y|z) \leftarrow \mathcal{N}(y|ty^+ + (1-t)y^-, \sigma^2)$
6     *Sample $t \sim [0,1]$ and $y \sim p_t(\cdot|z)$*
7     $\theta \leftarrow \theta - \eta\nabla\mathcal{L}(\theta)$   *# from (8)*
8   **until** $v_\theta$ *converges*;

9   *# Inference*
10   *Sample $y \sim \pi_{\mathrm{ref}}(\cdot|x)$*
11   *Solve ODE (3) with initial condition $\phi_0(y) = y$ using $v_\theta$*
12   **return** $\phi_1(y)$

---

# B  Empirical Evidence for Using Reference Policy Instead of the True Marginal During Inference

We conduct experiments on the D4RL walker2d environment to compare the performance of the flow matching method using $\pi_{\mathrm{ref}}$ and $p_0$ as source distribution, respectively.

Table 5: Normalized results on MuJoCo datasets. Mean and standard deviation from 5 seeds are reported.

|  | Pretrained (BC) | PFM from $\pi_{\mathrm{ref}}$ | PFM from $p_0$ | Planning |
|---|---|---|---|---|
| walker2d-random-v2 | $1.47 \pm 0.1$ | $1.77 \pm 0.13$ | $\mathbf{2.03} \pm \mathbf{1.01}$ | $1.93 \pm 0.13$ |
| walker2d-medium-v2 | $60.35 \pm 18.16$ | $72.59 \pm 15.8$ | $\mathbf{80.58} \pm \mathbf{1.80}$ | $76.27 \pm 7.97$ |
| walker2d-expert-v2 | $108.62 \pm 0.39$ | $108.38 \pm 0.28$ | $\mathbf{109.19} \pm \mathbf{0.37}$ | $109.08 \pm 0.21$ |
| Average | $56.81$ | $60.91$ | $\mathbf{63.93}$ | $62.43$ |

To replicate the actual marginal distribution of less preferred samples $p_0$, we sample two trajectories from a given state $s_t$, and use the ground truth reward function to select the less preferred data (with lower reward). Note that this use of the environment reward is restricted in practice, as it assumes access to the preference model during inference. We also compare another baseline model, namely a planning model, that searches among the two generated trajectories $\tau^+, \tau^- \sim \pi_{\mathrm{ref}}(\cdot|s_t)$ and chooses an action sequence with higher environmental reward.

As can be seen in Table 5, the flow matching model with the source distribution matched to the actual marginal distribution $p_0$ (as in the training process) achieves the highest performance among all baselines, including a flow matching model that uses $\pi_{\mathrm{ref}}$ instead of $p_0$. However, we observe that the flow matching model with $\pi_{\mathrm{ref}}$ as the source distribution yields comparable results to the

model using $p_0$ as source. It is also worth noting that the flow matching model (with true $p_0$) achieves better performance even compared to the planning method, that explicitly uses a trajectory with higher reward. This suggests that using a flow matching for preference alignment can provide better alignment than simply conducting an exhaustive search.

## C  Proof of Theorems

In this section, we provide proof to the theorems in the paper. Throughout this section, we write $\mu$ as the reference policy instead of $\pi_{\text{ref}}$ for the ease of writing.

First, we prove Theorem 4.1, where we rewrite the theorem statement for completeness.

**Theorem C.1** (**Characterization of the Marginal**). *Suppose that a preference dataset $\mathcal{D} = \{(x, y^+, y^-)\}$ is collected from a pre-trained reference policy $\mu$, i.e., $y^+, y^- \sim \mu(\cdot|x)$, where the preference $y^+ > y^-$ is labeled with probability $\mathbb{P}(y^+ > y^-|x)$. Let $p_1$ denote the marginal distribution of the positively-labeled samples $y^+$. Then the marginal distribution $p_1$ obtained from the preference model $\mathbb{P}(y > y'|x)$ is an optimizer of the optimization problem*

$$p_1 = \underset{\pi}{\text{argmax}} \, \mathbb{E}_{y \sim \pi} \bigg( \log \mathbb{E}_{y' \sim \mu} \Big( \mathbb{P}(y > y') \Big) \bigg) - \mathbb{D}_{\text{KL}} \Big( \pi \big\| \mu \Big). \tag{17}$$

*Proof.* For simplicity, we drop the conditional dependence of $x$. Notice that the general RLHF optimization objective (11) attains a unique optimal solution

$$\pi^*(y) \propto \mu(y) \, \exp \left( \frac{1}{\beta} \mathbb{E}_{y' \sim \mu} \Big( \sigma^{-1}(\mathbb{P}(y > y')) \Big) \right) \tag{18}$$

As it is a widely known result, we defer the proof to prior works, *e.g.*, Azar et al. [2023]. Intuitively, comparing the above solution with the target distribution $p_1$ as rewritten in (7) allows us to formulate a similar objective to (11) which $p_1$ optimizes. Formally, we begin with considering the optimization objective in (17):

$$\underset{\pi}{\text{argmax}} \, \mathbb{E}_{y \sim \pi} \bigg( \log \mathbb{E}_{y' \sim \mu} \Big( \mathbb{P}(y > y') \Big) \bigg) - \mathbb{D}_{\text{KL}}(\pi \| \mu) \tag{19}$$

$$= \underset{\pi}{\text{argmax}} \, \mathbb{E}_{y \sim \pi} \left( \log \mathbb{E}_{y' \sim \mu} \Big( \mathbb{P}(y > y') \Big) - \log \frac{\pi(y)}{\mu(y)} \right) \tag{20}$$

$$= \underset{\pi}{\text{argmin}} \, \mathbb{E}_{y \sim \pi} \left( \log \frac{\pi(y)}{\mu(y)} - \log \mathbb{E}_{y' \sim \mu} \Big( \mathbb{P}(y > y') \Big) \right) \tag{21}$$

$$= \underset{\pi}{\text{argmin}} \, \left( \log \frac{\pi(y)}{\mu(y) \mathbb{E}_{y' \sim \mu} \big( \mathbb{P}(y > y') \big)} \right) \tag{22}$$

$$= \underset{\pi}{\text{argmin}} \, \left( \log \frac{\pi(y)}{\frac{1}{Z} \mu(y) \mathbb{E}_{y' \sim \mu} \big( \mathbb{P}(y > y') \big)} - \log Z \right) \tag{23}$$

where $Z = \int \mu(y) \mathbb{E}_{y' \sim \mu} \big( \mathbb{P}(y > y') \big) dy$ is the normalization constant. Notice that the distribution in the denominator is precisely the marginal distribution $p_1$ of our interest, which is indeed a valid distribution. Hence, the last optimization objective can be reorganized as follows.

$$\underset{\pi}{\text{argmin}} \, \mathbb{D}_{\text{KL}}(\pi \| p_1) - \log Z. \tag{24}$$

Since the $Z$ is a constant, the KL-divergence objective is minimized at 0 if and only if the two distributions $\pi$ and $p_1$ are identical, by the Gibbs' inequality. It follows that $p_1$ is the optimal solution of the optimization problemm (17), which completes the proof. $\square$

Next, we aim to prove Theorem 4.2. Before stating and proving the theorem, we note here that the $L^2$ assumptions of the pre-trained model $\pi_{\text{ref}}$ and the probability model $\mathbb{P}(y > y')$ is generally valid in practical domains. For instance, assuming $\pi_{\text{ref}}$ being trained on $L_2$ loss as in many practical domains, $\pi_{\text{ref}} \in L^2$ generally holds. Also, the preference model $\mathbb{P}$ is usually provided in a finite support, or modeled as Bradley-Terry model, which generally behaves well.

**Theorem C.2** (**Convergence of Iterative Method**). *Assume that $\mu \in L^2$, and $\mathbb{P}(y > y') \in L^2$. Consider an iterative update of the marginal distribution $p_1$ as follows.*

$$p_1^{(n)}(y) \propto p_1^{(n-1)}(y) \int p_1^{(n-1)}(y')\mathbb{P}(y > y')dy', \quad p_1^{(0)} = \mu, \tag{25}$$

*i.e., we iteratively compute the marginal distribution of more preferred samples from the update marginal distribution. Then, the limiting marginal distribution $p_1^{(\infty)}$ converges to the uniform distribution $U$ of points $y$ where the value $\mathbb{E}_{y' \sim \mu}(\mathbb{P}(y > y'))$ is the largest, i.e.,*

$$p_1^{(\infty)} \to U\left(\underset{y}{\arg\max} \, \mathbb{E}_{y' \sim \mu}(\mathbb{P}(y > y'))\right). \tag{26}$$

*Proof.* Let us denote by $\mathcal{T}$ the transformation applied at each step:

$$\mathcal{T}[p](y) = \frac{1}{Z_p}p(y) \int p(y')\mathbb{P}(y > y')dy', \tag{27}$$

where $Z_p = \int p(y) \int p(y')\mathbb{P}(y > y')dy'dy$ is the normalization constant. Then, the update rule in (25) can be written simply as $p_1^{(n)} = \mathcal{T}[p_1^{(n-1)}]$. We aim to show that this iterative procedure converges to a distribution that places uniform weight on the set of points $y$ where $\mathbb{E}_{y' \sim \mu}(\mathbb{P}(y > y^-))$ is maximized. Given a probability distribution $p$, Let us define the function $f_p$ as follows.

$$f_p(y) \triangleq \mathbb{E}_{y' \sim p}\left(\mathbb{P}(y > y')\right) = \int p(y')\mathbb{P}(y > y')dy'. \tag{28}$$

Observe that $\mathcal{T}[p](y)$ increases $p(y)$ proportionally to $f_p(y)$. Consequently, the regions where $f_p(y)$ is higher will see an amplification in probability mass relative to regions where $f_p(y)$ is lower. This amplification occurs iteratively, with higher $f_p(y)$ regions increasingly dominating the distribution.

Formally, we claim that the fixed-point iterator $\mathcal{T}$ is compact. Notice that the kernel $K(y, y') = p(y')\mathbb{P}(y > y')$ is bounded, provided that $\mathbb{P}(y > y')$ is bounded by 1, and that $p$ is a probability density, which integrates to 1. By Schur's test and the properties of Hilbert-Schmidt operators, *i.e.*,

$$\iint |K(y, y')|^2 dy dy' < \infty, \tag{29}$$

$\mathcal{T}$ can be shown to be a compact operator, from the square integrability assumptions.

Next, consider the behavior of $\mathcal{T}$ on any sequence of probability densities $\{p^{(n)}\}_{n=0}^{\infty}$. By the properties of compact operators in the space of continuous functions, any sequence $\{p^{(n)}\}$ has a convergent subsequence in the weak topology. Let $p^*$ be the limit of any convergent subsequence. The uniform boundedness of $K(y, y')$ and the compactness of $\mathcal{T}$ suggest that $\mathcal{T}[p^*] = p^*$, establishing that $p^*$ is a fixed point of $\mathcal{T}$. To determine the nature of $p^*$, observe that

$$\int p^*(y) \int p^*(y')\mathbb{P}(y > y')dy'dy = 1. \tag{30}$$

Since $\mathbb{P}(y > y')$ is maximized uniformly over $y$ when $\mathbb{E}_{y' \sim \mu}(\mathbb{P}(y > y'))$ is maximized, $p^*$ must concentrate its mass on the set where this expectation is maximized. Therefore, $p^*$ converges to a uniform distribution over the set

$$\arg\max_{y} \mathbb{E}_{y' \sim \mu}(\mathbb{P}(y > y')). \tag{31}$$

Formally, recall that from Theorem 4.1, the updated distribution $p_1^{(n)}$ is a solution to the following optimization problem:

$$p_1^{(n)} = \underset{\pi}{\arg\max} \, \Psi(\pi) := \mathbb{E}_{y \sim \pi}\left(\log \mathbb{E}_{y' \sim p_1^{(n-1)}}\left(\mathbb{P}(y > y')\right)\right) - \mathbb{D}_{\mathrm{KL}}\left(\pi \middle\| p_1^{(n-1)}\right). \tag{32}$$

Hence, we note that if any point $y$ not in this set were to have a positive probability under $p^*$, then $\mathcal{T}[p^*]$ would not be equal to $p^*$, due to the strict maximization condition, contradicting the fixed-point property. Thus, $p_1^{(\infty)}$ converges to the uniform distribution over the optimal set, as stated. We note here that because the space of probability densities is closed under the topology induced by the function space norm, $p^*$ should also a probability density, and ultimately the unique minima. $\quad\square$

# D   Experimental Details

In this section, we describe our experimental details.

**Resource** All experiments were conducted on a single Nvidia Titan RTX GPU and a single i9-10850K CPU core for each run. The time required varies by task, but as it only involves collecting preference datasets and learning the flow, it completes within a few hours.

## D.1   Conditional Image Generation

**Task** We employ the MNIST [LeCun et al., 1998] [2] dataset to evaluate PFM on a conditional image generation task.

**Preference Dataset** We utilize a pre-trained DCGAN [Radford et al., 2015] generator as $\pi_{\mathrm{ref}}$ and collect sample pairs from $\pi_{\mathrm{ref}}$ conditioned on the digit labels as contexts. To construct preference datasets, we assign preferences to sample pairs according to the softmax probabilities of the labels from a LeNet [LeCun et al., 1998].

**Baseline** We consider the pre-trained DCGAN generator ($\pi_{\mathrm{ref}}$), a RLHF fine-tuned model and a DPO fine-tuned model of $\pi_{\mathrm{ref}}$ as baselines. All methods are trained until convergence, and we report the normalized episodes returns with the standard deviation from 5 different random seeds.

## D.2   Conditional Text Generation

**Task** To evaluate PFM on the NLP domain, we adopt a controlled (positive) sentiment review generation task with the IMDB dataset [Maas et al., 2011].

**Preference dataset** As done in Rafailov et al. [2024b], to perform a controlled evaluation, we adopt the pre-trained sentiment classifier[3] as the preference annotator. The preference dataset is constructed from randomly selected pairs of moview reviews $y^+, y^-$ from the IMDB dataset, where the preference is obtained from the classifier logit probability $p(\mathrm{positive}|y^+) > p(\mathrm{positive}|y^-)$. We then train our PFM model on this preference dataset, to obtain the marginal distribution of the preferred (positive sentiment) review $p_1(y^+)$.

**Baseline** We consider the GPT-2 SFT model on the IMDB dataset ($\pi_{\mathrm{ref}}$) and a RLHF fine-tuned model of $\pi_{\mathrm{ref}}$ using PPO ($\pi_{\mathrm{PPO}}$) as baselines. We apply PFM to baseline models as an add-on module without fine-tuning. For the iterative PFM, we iteratively apply the initially obtained learned preference flow without collecting new data from the improved PFM model.

**Embedding for variable-length input** For our PFM framework to be applied to variable-length inputs, we employ a T5-based autoencoder[4] to obtain fixed-sized (1024-dimensional vector) embedding of the input texts, allowing us to work within the fixed-size latent space. Once the fixed-size embeddings $z^+$ and $z^-$ are obtained for each text sample $y^+$ and $y^-$, we learn the conditional flow using PFM from $z^-$ to $z^+$. During inference, we apply PFM to the latent embedding $z$ of the given input text $y$, and decode the improved latent embedding using the T5 decoder. We adopt the same U-Net architecture used in our MNIST experiment, where we reshape the 1024-dimensional vector into a two-dimensional (32, 32) image tensor. As shown in Table 6, PFM requires a much smaller number of parameters (around 1.2%) than RLHF or DPO which involve fine-tuning LLMs.

Table 6: Parameters required for training for each method. PFM only requires 1.2% parameters to be trained compared to naive approaches (RLHF, DPO, etc.), and still achieves better performance in preference alignment.

| GPT-2 (RLHF) | U-Net (PFM) |
|---|---|
| 124M | **1.5M** |

**Iterative PFM** For our iterative PFM, we do not iteratively collect data and re-train the module. Instead, we simply iteratively apply the learned preference flow to the output samples. In particular, we apply the learned flow to the embedding $z$ iteratively with the same flow module.

---

[2] http://yann.lecun.com/exdb/mnist

[3] https://huggingface.co/lvwerra/distilbert-imdb

[4] https://huggingface.co/thesephist/contra-bottleneck-t5-large-wikipedia

**Prompt for GPT-4 Evaluation** Below we include the prompts used to generate win rates.

```
Which of the two movie reviews has a more positive sentiment?
Response A: <Response A>
Response B: <Response B>
```

### D.3 Offline Reinforcement Learning

**Task** To assess the performance of PFM in reinforcement learning tasks, we employ the D4RL [Fu et al., 2020] benchmark from `https://github.com/Farama-Foundation/D4RL` where the datasets and code are licensed under the Creative Commons Attribution 4.0 License (CC BY) and the Apache 2.0 License, respectively. We consider four different tasks (`ant`, `hopper`, `halfcheetah`, `walker2d`) from Gym-Mujoco domain with three different levels of offline dataset quality (`random`, `medium`, `expert`) for each task.

**Preference dataset** We first pre-train $\pi_{\text{ref}}$ using behavior cloning (BC) for each offline dataset. We then collect segment pairs of the rollout trajectories from $\pi_{\text{ref}}$, with each pair starting from the same state as a context. To construct preference datasets, we utilize a scripted teacher which prioritizes trajectories with higher rewards based on the ground truth rewards provided by the environment. This approach has been widely adopted in the PbRL settings [Kim et al., 2023, Lee et al., 2021, Zhu et al., 2024]. The preference datasets consist of 1,000 pairs of preferred and rejected segments and their context for each offline dataset, with the segment length 10.

**Baseline** The baseline methods for comparing the performance of PFM includes behavior cloning (BC), which we adopt as our pretrained reference model $\pi_{\text{ref}}$, and a DPO fine-tuned model from the BC model. For DPO fine-tuned models, we search KL regularization coefficient $\beta$ from 0.01 to 100 and adopt the best one. Additionally, we train a separate behavior cloning model to the collected preferred samples $y^+ \sim p_1$, aiming to replicate the marginal distribution of the "good" trajectories. All methods are trained until convergence, and we report the normalized episodes returns with the standard deviation from 5 different random seeds.

## E  Broader Impacts

As an add-on module, PFM can be seamlessly integrated into various real-world AI applications, such as generative models and continuous control systems, without the need to modify the underlying application models. This integration enables the applications to deliver personalized results that align with individual user preferences. However, since PFM utilizes preference data, it raises potential privacy concerns similar to those found in typical PbRL methods. These concerns can be mitigated by ensuring that access to user preferences is granted only with explicit user consent.

