# OpenReview forum: "Preference Alignment with Flow Matching"
_NeurIPS.cc/2024/Conference — NeurIPS 2024 poster_

### Official Review · Reviewer_XK5w · 2024-06-13

**Soundness:** 3
**Presentation:** 3
**Contribution:** 1
**Rating:** 3
**Confidence:** 4

**Summary:**

The paper introduces a novel framework called Preference Flow Matching (PFM) for preference-based reinforcement learning (PbRL). The PFM approach aims to integrate human preferences into pre-trained models without the need for extensive fine-tuning, which is a common requirement in existing PbRL methods. The PFM framework utilizes flow matching techniques to learn directly from preference data, reducing the dependency on pre-trained model fine-tuning. It transforms less preferred data into more preferred outcomes by leveraging flow-based models, aligning model outputs with human preferences without explicit reward function estimation. The paper provides theoretical insights and empirical results that demonstrate the effectiveness of PFM in aligning pre-trained models with preferences and offers a new direction in PbRL. PFM is proposed as a solution to challenges such as scalability, inefficiency, and the need for model modifications, especially with black-box APIs like GPT-4.

**Strengths:**

Innovative Approach: PFM offers a fresh perspective on integrating human preferences into AI systems by using flow matching, which is a relatively under-explored method in preference alignment.

Theoretical Foundation: The paper provides a solid theoretical basis for PFM, proving its alignment with standard PbRL objectives and robustness against overfitting in reward models.

Empirical Validation: Experimental results support the practical effectiveness of PFM, showing its ability to robustly align with preferences and achieve comparable performance to traditional RLHF methods and DPO.

Scalability and Efficiency: By eliminating the need for fine-tuning large pre-trained models, PFM addresses significant challenges related to scalability and computational efficiency.

Applicability to Black-Box Models: PFM's ability to work as an add-on module for black-box models, like GPT-4, extends its applicability to scenarios where model access is limited.

**Weaknesses:**

Exaggerated Motivation:  I believe the research motivation stated by the author in the abstract and introduction may be exaggerated. For example, the author claims that the method can be applied to black-box models like GPT-4, but there is no experimental validation. In fact, even for general language models, the author provides no experimental evidence to show that the proposed method is superior to DPO.

Limited Domain Testing: The paper's experiments may not fully explore the potential of PFM across diverse domains, particularly in more complex tasks like general contextual generation.

Assumption is too strong : The assumption is not general for RLHF scene. For example,  conditional flow is $y^+ - y^-$ (line 116), and the  probability path is subject to gaussian distribution,  which is not pratical for general AI model, such as GPT-4.

Potential for Overfitting: Although PFM is designed to be robust against overfitting, the method might still be susceptible to overfitting in certain scenarios, particularly if the preference data is limited or biased.

Applicability to NLP Tasks: The current design of PFM may not be directly applicable to natural language processing tasks due to challenges with variable-length data, which could be a significant limitation given the prevalence of NLP applications.

**Questions:**

1. I believe the assumptions in the paper are overly idealistic, as the method appears impractical for application to black-box models like GPT-4, and may even struggle with open-source models. If the authors could provide a specific plan on how to apply the method to open-source models (such as llama) and black-box models (such as GPT-4), including details of experiments and experimental results, I would consider raising the score. Otherwise, we suspect the research motivation in the paper has been exaggerated.

2. How to apply the simple assumptions from lines 116-117 to LLM?

**Limitations:**

The authors have made an effort to address the limitations and potential negative societal impacts of their work.

---

> ### Author Rebuttal · Authors · 2024-08-04
>
> Dear reviewer XK5w, thank you for your careful review and constructive feedback. Our responses to each of your comments are presented below.
>
> - **W1: Exaggerated motivation, limited domain testing and applicability to NLP tasks.**
> Thank you for pointing out this concern. We also acknowledge the importance of further evaluation of PFM, especially on NLP domain. To address this issue, we conducted additional experiments on the NLP task. Please refer to our common response.
>     - We applied PFM to the open-source LLM (GPT-2) for text generation task.
>     - We designed a pipeline to use PFM for variable-length text data.
>     - We compared PFM with RLHF fine-tuned model.
> - **W2: Too strong assumption on conditional flow.**
> We follow the unified framework for conditional flow matching (Tong, Alexander, et al. "Improving and generalizing flow-based generative models with minibatch optimal transport." *Transactions on Machine Learning Research*). We highlight that the Gaussian probability path is not a required assumption, rather it is a design choice. Following the original work of the CFM, we define the probability path between $y^+$ and $y^-$ as gaussian distribution for ease of usage and theoretical guarantee. Hence this does not hinder the applicability of PFM. All assumptions required for our framework to be theoretically guaranteed are thoroughly discussed in the Appendix A and C of our paper. We note here that these assumptions are generally valid across various ML domains.
> - **W3: Potential for Overfitting.**
> If the preference data is limited or biased, preference-based methods may generally suffer from overfitting issue. This limitation is not restricted to our method; we believe that overfitting is generally inevitable in case of limited data for any existing ML frameworks.
> - **Q1: Overly idealistic motivation.**
> Please refer to our common response. We applied PFM to the open-source LLM (GPT-2) for text generation task, included details of experiments, and reported the results compared with SFT and RLHF models.
> - **Q2: How to apply the simple assumptions from lines 116-117 to LLM?**
> In our additional experiments included in our global response, we embedded text data to fixed-size latent vectors and learned the vector field for the probability path in the latent space.

---

### Official Review · Reviewer_2tji · 2024-06-28

**Soundness:** 2
**Presentation:** 3
**Contribution:** 2
**Rating:** 4
**Confidence:** 4

**Summary:**

This paper presents Preference Flow Matching (PFM), which learns an ODE to transform less-preferred data into more-preferred ones. Due to PFM not explicitly or implicitly relying on reward functions, it avoids reward overfitting. Empirical evidence demonstrates that PFM outperforms DPO in image conditional generation and Offline RL tasks.

**Strengths:**

- The paper is well-written, with illustrative figures and toy examples that are intuitive and easy to understand.
- The approach of using a flow model to transform less-preferred data into more preferred ones is straightforward and effective.

**Weaknesses:**

- The novelty of the paper appears somewhat limited, as PFM seems to use Conditional Flow Matching (CFM) to transform between less and more preferred data distributions. However, the paper does not thoroughly discuss the benefits of introducing CFM. Similar transformations can be achieved with Rectified Flow or Diffusion models, and work using Diffusion models (such as FTB) already exists.
- The experimental validation in the paper is not sufficiently comprehensive, leading to less convincing results. For instance, in the context of Offline RL tasks, the following issues may exist:
    - PbRL is commonly used to address tasks with difficult-to-define or sparse rewards. In D4RL, tasks like antmaze are more commonly used for better algorithm performance comparison.
    - In MuJoCo tasks, PFM does not exhibit a significant improvement over DPO. Additionally, the standard deviation of the results appears to be quite large, making comparisons based solely on average means less reliable. Could this be due to limited random seeds? Typically, MuJoCo tasks have low standard deviations for having dense rewards.
    - While the authors summarize many RLHF methods in the related works section, only DPO and vanilla RLHF are compared across the experiments, indicating a potential lack of baseline algorithms. For example, FTB uses a diffusion model to do similar tasks as PFM. Given the similarity between these two algorithms, why are they not compared in the study?

**Questions:**

Please see Weaknesses.

**Limitations:**

Please see Weaknesses.

The main areas for improvement in the paper are concentrated in the experimental section, as the current experimental results lack persuasiveness.

---

> ### Author Rebuttal · Authors · 2024-08-04
>
> Dear reviewer 2tji, thank you for your careful review and valuable comments. Our responses are presented below.
>
> - **W1: Main novelty of our method.**
> The main novelty of our method comes from its capability of being added on top of any existing black-box models, without the need of fine-tuning this reference policy class. FTB for instance, requires an additional policy extraction phase, as FTB only focuses on the data augmentation for the behavior cloning policy. This means that FTB also requires a fine-tuning of the given reference policy in order to obtain the final policy.
> - **W1-1: Choice of conditional flow matching (CFM)**
> Regarding the choice of conditional flow matching (CFM), our method can indeed be extended to frameworks that use rectified flow and diffusion models. As mentioned in the CFM paper (Tong, Alexander, et al. "Improving and generalizing flow-based generative models with minibatch optimal transport." *Transactions on Machine Learning Research*), rectified flow and conditional flow matching can be unified under a single flow-matching framework. The key difference of the rectified flow and the CFM only comes from the design choice of the flow transport model; CFM chooses a Gaussian probability path, whereas the rectified flow chooses a Direc probability path for density transportation. We would like to emphasize that the novelty of our approach comes from the probability transport module to be added on top of the existing black-box reference policy. To the best of our knowledge, we are the first to adopt such novel fine-tuning-free methods to improve alignment with the preference. This approach will be more favorable in cases where the fine-tuning of the original reference policy is unavailable, or computationally expensive.
> - **W2: Weak empirical results.**
> Although PFM does not gain significant improvement over DPO in D4RL benchmarks, it is important to note that we have gained a similar performance increase from the reference policy, without any fine-tuning of the original policy. That being said, the greatest advantage of our method is its ability to be attached to any black-box model for improved preference alignment. This is verified across variable domains: vision, RL, and even language domains. Please refer to the common response above for our newly added experimental results for the NLP task. We believe these results will serve as strong enough evidence that our method can be used to improve preference alignment.
> - **W2-1: Regarding D4RL results**
> As you pointed out, we observe high variance across all the results in the MuJoCo experiments. However, this is not due to limited random seeds or insufficient experiments. In offline RL tasks, compared to online RL settings, the variance may be large due to its limited access to the environment interaction. Especially in PbRL settings, the variance may be larger due to the lack of available preference-labeled data. This large variance can be observed in some other PbRL papers too; for instance, see Preference Transformer (Kim, Changyeon, et al. "Preference transformer: Modeling human preferences using transformers for rl.") for a similar scale of variance.
> - **W2-2: comparing our method to existing frameworks.**
> As mentioned previously, we are the first to apply a fine-tuning-free method to improve preference alignment. Hence, our experiments mainly focused on comparing our new framework to the existing standard frameworks like RLHF and DPO. In the case of FTB, this method also requires an additional policy extraction phase that fine-tunes the reference policy. Furthermore, FTB suffers from heavy computational costs both for the policy extraction and the data augmentation using diffusion models. Therefore, we believe including FTB as a baseline is unfair, as it trains on a larger augmented dataset. Of course, our method can be applied concurrently with the FTB; FTB can be used to augment more quality datasets, where PFM can then be applied on top of the augmented preference dataset for additional performance gain.

---

> > ### Comment · Reviewer_2tji · 2024-08-08
> >
> > Thank you very much for the thorough response. I still have some unanswered questions. I provide more details below:
> > - What metric is used in Table 1? To my knowledge, the standard D4RL normalized score differs significantly in magnitude from the results presented in Table 1. I excerpt the D4RL scores for PFM, FTB, PT, and BC in the table below. The results for FTB and PT are sourced from the FTB paper's report [1], while the BC results are from the D4RL report [2]. Except for BC, all three methods utilized script teachers. PFM's results differ significantly in magnitude. So I wonder if PFM used a different metric. However, the appendix only mentions "we report the normalized episodes returns." My mention in the review that "the standard deviation of the results appears to be quite large" also refers to the discrepancy in magnitude between the standard deviation and mean values. It is understood that offline PbRL would naturally exhibit a notably larger standard deviation, but comparing it to the mean score in terms of magnitude is also reasonable (see results of FTB and PT).
> >
> > |                         | PFM (Yours) | FTB         | PT (Preference Transformer) | BC   |
> > | ----------------------- | ----------- | ----------- | --------------------------- | ---- |
> > | hopper-medium-expert-v2 | -           | 111.1 ± 2.0 | 77.8                        | 52.5 |
> > | hopper-medium-replay-v2 | -           | 89.6 ± 4.9  | 31.4                        | 18.1 |
> > | hopper-expert-v2        | 3.6 ± 0.8   | -           | -                           | -    |
> > | hopper-medium-v2        | 1.9 ± 2.7   | -           | -                           | 52.9 |
> >
> > - It is indeed a default setting in FTB to employ an additional policy extraction phase. However, FTB can also, like PFM, directly enhance the quality of action trajectories during inference. This means that FTB can also operate on black-box models. In this context, if we use FTB in this manner, what are the main contributions of PFM compared to FTB? (I am not challenging the authors' work; I simply seek a better understanding.) Additionally, I do not consider the policy extraction phase in FTB to be very costly, nor do I view it as a drawback. The BC policy used in FTB's policy extraction phase can be a very small MLP that does not require significant computational resources. Moreover, during inference, it avoids calling the generative model for trajectory optimization at each step, making it faster and more lightweight.
> >
> > ---
> > References
> >
> > [1] Zhilong Zhang, et al, "Flow to Better: Offline Preference-based Reinforcement Learning via Preferred Trajectory Generation," in The Twelfth International Conference on Learning Representations, 2024.
> >
> > [2] Justin Fu, et al, "D4RL: Datasets for Deep Data-Driven Reinforcement Learning," in arXiv, 2004.07219, 2021.

---

> > > ### Author Response · Authors · 2024-08-08
> > > **Clarification of the D4RL Results.**
> > >
> > > Thank you for your detailed feedback and for raising these important questions. We appreciate the opportunity to further elucidate the contributions and distinctions of our framework, and the opportunity to clarify the metrics used in our paper, particularly in Table 1.
> > >
> > > First, let us clarify the metrics reported in Table 1. The standard D4RL normalized score is calculated using the formula:
> > >
> > > $\text{normalized score} = 100 \times \frac{\text{score} - \text{random score}}{\text{expert score} - \text{random score}} $
> > >
> > > In other words, they report the score for the random policy as 0, and the score for the optimal (or expert) policy as 100. While the FTB paper uses this D4RL normalized score, in our work, we opted to report the cumulative rewards of each method (referred to as un-normalized scores in the D4RL paper, Table 3) normalized to a scale of 1000 for better readability. This choice was made to provide a clear and direct comparison of the methods' performance. Accordingly, in our normalization process, we divided the standard deviation by $\sqrt{1000}$, resulting in the mean and standard deviation appearing similar in scale. This method was chosen for consistency and to maintain a straightforward interpretation of the results. We would like to note that this choice also allows us to better compare the performance levels with respect to the reference policy that each method started with, not with the ground-truth worst and best policy as standards. Below, we provide the results of Table 1 obtained via the D4RL normalization scheme as reference:
> > >
> > > | Environment            | DPO Fine-tuned    | PFM (Ours)       |
> > > |------------------------|-------------------|------------------|
> > > | ant-random-v2          | -0.1 ± 0.1        | **0.0** ± 0.1    |
> > > | ant-medium-v2          | 67.3 ± 14.8       | **69.1** ± 2.6       |
> > > | ant-expert-v2          | **109.7** ± 4.0   | 106.8 ± 2.9      |
> > > | hopper-random-v2       | 0.1 ± 0.2         | **4.2** ± 0.1    |
> > > | hopper-medium-v2       | 46.5 ± 3.6        | **51.4** ± 2.4   |
> > > | hopper-expert-v2       | 100.2 ± 0.9       | **100.4** ± 0.7  |
> > > | halfcheetah-random-v2  | **0.0** ± 0.0     | **0.0** ± 0.0    |
> > > | halfcheetah-medium-v2  | 44.7 ± 0.8        | **46.5** ± 1.0   |
> > > | halfcheetah-expert-v2  | **101.3** ± 0.9   | 98.9 ± 0.9       |
> > > | walker2d-random-v2     | -0.1 ± 0.1        | **0.3** ± 0.1        |
> > > | walker2d-medium-v2     | **67.7** ± 11.3   | 66.4 ± 14.7      |
> > > | walker2d-expert-v2     | **99.8** ± 0.3    | **99.8** ± 0.2   |
> > >
> > > Here, we used the scores of BC-random and BC-expert in place of the scores of actual random and expert policies for normalization. (That is, the pretrained-BC models are set to 0 and 100, for random and expert, respectively.) Although this is not the exact same normalization scheme as the standard D4RL benchmarks, since both methods transform the scores to the same scale, the results remain consistent.

---

> ### Author Response · Authors · 2024-08-08
> **Key Contributions and Differences of PFM Compared to FTB**
>
> To address your second question and provide a clearer understanding of the differences and contributions of PFM compared to FTB, we offer the following explanation:
>
> Firstly, let us revisit FTB and its approach. FTB creates $K$ blocks of preference levels. Since the labeled or unlabeled datasets are not initially clustered into these $K$ levels, FTB requires score function estimation to assign trajectories to each block (or cluster). Once the dataset is classified into these $K$ performance-based blocks, FTB learns a diffusion model that transports a trajectory to match the score level of the neighboring better block. According to FTB, this transition between neighboring blocks ensures that the performance gap between each pair of trajectories is relatively consistent. Notably, FTB chose $K=20$ as the number of blocks and experimented with a minimum of 10 blocks even in their ablation study. The key motivation of FTB is to *unconditionally* generate good-quality trajectories within the fine-grained blocks. Note that this approach does not condition on the current state, but on the entire trajectory, making direct inference application challenging without additional steps. They also rely on the relatively close distance between these trajectory samples, as they do not condition on the same initial state, but simply optimizes for the complete trajectories, unconditionally. In short, FTB focuses on obtaining better trajectories, but not necessarily better trajectory *for the current starting state*.
>
> In contrast, PFM aims to improve the trajectories generated by a reference policy in a manner that is immediately usable during inference. In other words, we aim to directly find the trajectory (or improve upon the current trajectory) that will lead to better results given the current state and situations. This is achieved by learning a *vector field* that directly enhances the trajectory according to current conditions (e.g. current initial state condition). Unlike FTB, which improves trajectories unconditionally within predefined score blocks, PFM conditions on the current state, making direct inference application more straightforward without additional steps. Notably, once we obtain a vector field that characterizes the probability transport of trajectories, we can iteratively apply this learned vector field to achieve the same improvement effect as FTB’s iterative upgrade to the neighboring higher-scored block. Our flow model improves the trajectory while preserving the conditional information of the current situation, making our method more interpretable and adaptable.
>
> If one reduces FTB to having a block number $K=2$ and removes the policy extraction phase, FTB's approach becomes approximately similar to PFM, as you suggested. However, as mentioned above, the core motivations differ significantly. The authors of FTB did not reduce the block number to 2 (as in standard RLHF frameworks) and added a policy extraction phase due to the inability of FTB to be directly applied during inference, as its flow does not conditionally improve the trajectory.
>
> Another primary benefit of PFM is its ability to enhance action trajectories without the need for score functions. PFM avoids overfitting in reward models by directly aligning model outputs with human preferences through flow-based models. This approach contrasts with FTB’s reliance on score functions for block assignment, which can lead to overfitting and suboptimal performance, as mentioned in our paper. PFM’s method of learning from preference data directly addresses these concerns, providing a more robust solution.
>
> While FTB’s unconditional data augmentation approach can be beneficial in certain reinforcement learning (RL) scenarios, it is less adaptable to other tasks, such as language modeling. In language tasks, generating contextually appropriate sentences is crucial. PFM’s conditioning on the current state allows for more relevant and context-specific trajectory improvements, making it more versatile across various applications. FTB, on the other hand, focuses on generating high-return sentence sets, which is less practical for tasks requiring specific contextual appropriateness. Furthermore, directly applying FTB to language tasks requires an additional SFT phase to the obtained augmented dataset.
>
> In summary, PFM provides a more efficient and versatile method that is particularly well-suited to black-box models and diverse applications beyond RL. By learning a vector field conditioned on the current state, PFM can directly enhance trajectories in real-time, making it a valuable contribution to general preference alignment in diverse domains. We hope this explanation clarifies the unique aspects and contributions of PFM. We appreciate your insightful questions and are happy to engage in further discussion to deepen the understanding of our work.

---

> > ### Comment · Reviewer_2tji · 2024-08-12
> >
> > Thank you for your response. However, I have to say that it does not fully address my concerns. I still have the following questions:
> >
> > - Why does the author still not use the exact D4RL score in the new Table 1? The author only needs to call `env.get_normalized_score` without training new models or re-testing existing ones. What's the rationale behind training a new BC-random and a new BC-expert for normalization? This undoubtedly makes fair comparisons difficult. Moreover, the reported scores in new table don't seem competitive compared to other Offline PbRL algorithms with script teachers. I suspect the author may not have thoroughly tuned the implementations. To verify this, I implemented a DPO myself. The policy is a squashed Gaussian distribution, using a 3-layer MLP with 256 hidden units. Following the same procedure as in Appendix D.2, I obtained scores on Hopper tasks that are significantly better than those reported by the author (see table below). This might be due to my use of very small beta values: 1e-10 for expert, 1e-8 for medium, and 5e-9 for random, fine-tuned over 500 gradient steps with a batch size of 32. Regardless, I believe PFM's current performance on D4RL-MuJoCo is not satisfactory.
> >
> > |env|my DPO| author's DPO|
> > |---|---|---|
> > |hopper-expert|110.7+-0.7|100.2+-0.9|
> > |hopper-medium|79.7+-1.2|46.5+-3.6|
> > |hopper-random|7.6+-0.1|0.1+-0.2|
> >
> > - While implementing DPO, I noticed that PFM requires online interaction with the environment when collecting the preference dataset. This seems entirely inappropriate for an offline setting. One of the main challenges in offline settings is the limited dataset, and introducing online interaction data would significantly alleviate the OOD issue. I believe this approach may introduce unfair comparisons. Additionally, PFM requires interaction with the environment during inference to obtain a reference trajectory and then optimizes it, which assumes access to correct dynamics. If this is the case, why not directly use a preference-based reward function for planning? At the very least, I think this would be a reasonable baseline to include.
> >
> > - I believe the experimental section is currently insufficient, a point also raised by other reviewers. Firstly, the chosen benchmarks are relatively simple: MNIST for image generation and only MuJoCo for PbRL. The author's analysis of the experimental results lacks depth, seemingly offering not enough insightful takeaways beyond "PFM has better performance." Also, I suggest the author include more baselines, such as IPO and FTB. These methods are compared in the related works section but not in the experiments, especially FTB, which is most similar to PFM.
> >
> > - I strongly agree with reviewer XK5w's point that while the author claims the method can be applied to black-box models like GPT-4, there's no experimental validation of this. More experiments on black-box models would better showcase PFM's advantages and support the author's claims. Why didn't the author choose to include experiments on GPT-4 in the rebuttal? If cost is a concern, even experiments with GPT-3.5 would help demonstrate PFM's strengths.
> >
> > Given these unresolved issues, I will maintain my current rating.

---

> > > ### Author Response · Authors · 2024-08-13
> > > **Additional Response to Reviewer 2tji**
> > >
> > > Thank you again for your invaluable comments. Below, we address your additional concerns:
> > >
> > > **Regarding the D4RL experimental results.**
> > >
> > > Thank you for addressing the appropriate normalization scheme for D4RL tasks. Below, we refine our results based on the call `env.get_normalized_score` on our trained models.
> > >
> > > Environment|BC|DPO Fine-tuned|PFM (Ours)|Marginal BC
> > > -|-|-|-|-
> > > ant-random|31.59±0.05|31.52±0.08|**31.62±0.13**|25.01±4.64
> > > ant-medium|90.16±21.48|95.04±13.93|96.73±2.47|**99.67±1.57**
> > > ant-expert|125.83±24.07|**134.96±3.76**|132.2±2.69|99.29±34.74
> > > hopper-random|3.17±0.25|3.23±0.25|**7.69±0.08**|5.48±4.46
> > > hopper-medium|52.83±5.03|53.47±3.92|**58.76±2.62**|40.44±1.69
> > > hopper-expert|111.27±0.48|111.51±0.92|**111.7±0.77**|32.39±0.1
> > > halfcheetah-random|2.25±0.01|2.26±0.01|**2.26±0.0**|2.21±0.02
> > > halfcheetah-medium|40.97±0.89|41.94±0.68|**43.49±0.88**|38.79±1.27
> > > halfcheetah-expert|91.02±1.24|**92.15±0.76**|90.05±0.83|4.77±2.5
> > > walker2d-random|1.47±0.1|1.38±0.08|1.77±0.13|**2.45±0.38**
> > > walker2d-medium|60.35±18.16|**74.05±12.05**|72.59±15.8|65.29±12.58
> > > walker2d-expert|**108.62±0.39**|108.38±0.28|108.36±0.21|15.8±0.54
> > >
> > > Please notice that the scale is now close to the values that you reported. We note that our comparison with DPO is fair since we trained on the same dataset, using the same policy network. The reason your reported values might be slightly different from our results include:
> > > - Different datasets used to train your DPO and ours.
> > > - Policy network used for both DPO and PFM for our experiments is slightly different from yours; we adopted the same architecture used in the DDPG implementation of D4RL repository.
> > >
> > > Nonetheless, we believe that our D4RL experiments were conducted fairly, and our current experimental results are adequate. Additionally, our results are comparable to DPO, if not slightly better in some cases, indicating that our method could be viable for black-box settings where DPO may not be applicable. We've also provided comprehensive results across various tasks and domains, including the NLP results mentioned in the common response section. We believe that these results sufficiently demonstrate the general applicability and effectiveness of our framework.
> > >
> > > **Data collection for PFM.**
> > >
> > > In the case of PbRL tasks, we conducted all the experiments on a pre-collected dataset from the behavior policy and trained policies on this data in an offline setting. Therefore, contrary to your concern, we did not violate the offline setting. Of course, it is true that we collected data separately, rather than using the existing offline dataset, in order to obtain trajectories starting from the same state. However, this is a common issue to be faced by all methods that make inferences from the same context (in the case of RL tasks, the initial state), not just by PFM. The assumption of PFM is to align preferences for a given context, so we collected a dataset to best demonstrate this setting. Please note that in other domains (e.g., vision, language), data is generally provided in this form, so this issue only arises in RL tasks.
> > >
> > > **Empirical results for PFM on various tasks.**
> > >
> > > We believe that our experimental results are sufficient to demonstrate the value of PFM. We conducted additional experiments not only in the RL and vision domains, but also on NLP tasks. In all of these tasks, our method performed comparably to baselines or, at times, even better. As mentioned above, we believe this is sufficient to show that our method has the potential to function as a new framework. Furthermore, our method is based on a theoretical guarantee, which is also one of our main contributions.
> > >
> > > **Regarding additional baselines.**
> > >
> > > For RL tasks, we plan to include FTB and IPO as additional baselines. We are currently working with the code, but due to time constraints and the challenge of ensuring a fair comparison between PFM and FTB under the same conditions, we were unable to include these results during the discussion period. However, we will make an effort to include these baseline results by the camera-ready deadline. Furthermore, we will also try to implement the preference-based reward function planning that you suggested, as an additional baseline.
> > >
> > > **Regarding NLP results.**
> > >
> > > We have provided results on an NLP task in the common response to support our claim that PFM can be applied to black-box models. Specifically, we used GPT-2 as a baseline in our experiments, where we treated the GPT-2 as a black-box model in our experiments. The reason we chose GPT-2 instead of GPT-4 is that the publicly available SFT model trained on the IMDB task, as well as the RLHF fine-tuned model, are both based on GPT-2. While our method could be applied to GPT-4, a fair comparison with the RLHF method isn't possible because no GPT-4-based RLHF fine-tuned model is available. Additionally, it's worth noting that the DPO paper also used the same GPT-2 base model for performance comparisons.

---

> > > > ### Comment · Reviewer_2tji · 2024-08-13
> > > >
> > > > Thank you for your response. I greatly appreciate the authors' efforts. Here's my reply:
> > > >
> > > > **Regarding the D4RL experimental results**
> > > > The new table's score report is much more readable. However, I still have some doubts about the DPO results. From the table, it seems that DPO fine-tuning has almost no effect in many environments. In my implementation, though, I saw noticeable improvements with low beta values and few-step updates. I used a very simple MLP neural network structure, so the performance difference might be due to different beta values, or perhaps because I used a squashed Gaussian distribution, which may be more robust. Anyway, I believe using standard D4RL scores improves readability, and I suggest the authors adopt this approach in their revisions.
> > > >
> > > > **Data collection for PFM**
> > > > I still maintain that offline PbRL settings shouldn't allow additional online interaction data collection. However, since the authors used the same training method for PFM and all baselines in their algorithm comparison, I think this fairly reflects the algorithms' performance in preference matching. So this may not be an issue after all.
> > > >
> > > > **Empirical results for PFM on various tasks / Regarding additional baselines / Regarding NLP results**
> > > > I still believe the current testing tasks are insufficient. I appreciate that the authors conducted experiments across RL, CV, and NLP domains, demonstrating PFM's versatility, but the tasks used in these three domains are too simple (MuJoCo, MNIST, IMDB). Regarding the experimental setup, I have the following suggestions:
> > > >
> > > > * Add IPO and FTB as baselines. For RL tasks, I believe the MPC baseline is unnecessary, as mentioned in the "Data collection for PFM" part.
> > > > * For RL tasks, try medium-expert and medium-replay datasets. For PbRL (or RLHF), clear contrasts are important. The currently chosen random/medium/expert datasets are collected from policies with uniform performance, making comparisons difficult and potentially obscuring the algorithm's full capabilities.
> > > > * For RL tasks, try antmaze datasets. These provide diverse datasets and use extremely sparse reward functions, which is where PbRL can show its capability.
> > > > * For all tasks, add real human feedback experiments. Almost PbRL or RLHF papers include experiments with real human feedback on typical tasks to better compare algorithm performance in practical applications. The current script teacher experiments may have some reward hacking issues - for example, in Figure 3, I think (d) PFM is clearly more recognizable than (h) Iterative PFM.
> > > > * For NLP tasks, add experiments with black-box models like GPT-3.5 or GPT-4. Sorry if I was unclear before - I understand that black-box models can't be fine-tuned using DPO/RLHF. What I meant was that GPT-4 could provide good real human feedback experiments. For instance, train a PFM on a small text dataset to make sentences less friendly, then have humans evaluate whether PFM can make GPT-4's output less friendly. It's known that GPT-4 is fine-tuned to reduce unfriendly content generation, so if PFM could change this, it would be very interesting.
> > > >
> > > > All in all, I still think the paper's experimental content is insufficient, so I'll maintain a negative rating. However, I appreciate the authors' efforts in the rebuttal and our productive discussion. I look forward to the additional baselines being added, so I'll increase my rating by 1.
> > > >
> > > > Also, this paper's ratings currently show a high variance. As the discussion period is ending, I've noticed that I seem to be the only reviewer discussing with the authors. I strongly recommend that the authors remind (or ask the AC to remind) other reviewers who haven't participated in the discussion to share their opinions. Thank you again for your response.

---

> > > > > ### Author Response · Authors · 2024-08-14
> > > > >
> > > > > Thank you so much for your valuable feedback and the productive discussions. We appreciate your detailed comments that helped us improve our research.
> > > > >
> > > > > As you suggested, we are working on further improving our experimental setup. Specifically, we are planning to conduct additional experiments in datasets such as medium-replay and medium-expert, with additional baselines. We also intend to obtain more results for NLP tasks.
> > > > >
> > > > > One of the key advantages of PFM is its ability to consistently enhance preference alignment as an add-on method, regardless of the given black-box reference policy. We have demonstrated this theoretically. To further emphasize this strength, we will include a wider range of reference policies in our additional experiments. Particularly for the D4RL benchmarks, we aim to employ various baselines and incorporate more reference policies to showcase that PFM is a versatile framework, capable of being applied across diverse settings and black-box models. We thank you again for your time and effort.

---

### Official Review · Reviewer_XhGj · 2024-07-12

**Soundness:** 3
**Presentation:** 3
**Contribution:** 2
**Rating:** 6
**Confidence:** 3

**Summary:**

This paper proposes an add-on method named Preference Flow Matching (PFM) to achieve preference alignment without learning a reward model. PFM can transfer $y$ generated from the original model to the preferred $y^+$ through a few flow iterations. Experiments demonstrate its superior performance compared with RLHF and DPO.

**Strengths:**

-This paper is well-written. I really enjoy reading it.
- While flow matching is not novel, this paper proposes to adapt it to the preference alignment area and yields many advantages (e.g., simplicity and efficiency).
- Experiments on both image generation and offline RL demonstrate its comparable performance.

**Weaknesses:**

- During inference, PFM with prior as $π_{\rm ref}$ suffers from a shifted source distribution, which can affect the performance.
- Limited testing benchmarks. RLHF is getting a lot of attention these days, largely due to its success in the language model area. However, this paper does not include the language-based task because PFM is limited in its ability to generate variable text.

**Questions:**

- During inference on the D4RL tasks, how do you choose actions to interact with the environments? As the paper claims, PFM samples an action trajectory $\tau$ from $π_{\rm ref} (·|s_t)$, and then applies flow matching to obtain a better action sequence $\tau_{\rm new}$. How can you sample a trajectory $\tau$ without training a dynamic model? Do you apply all actions in $\tau_{\rm new}$ to interact with the environment, or select some actions from $\tau_{\rm new}$ to take?

**Limitations:**

Limitations have been discussed in this paper.

---

> ### Author Rebuttal · Authors · 2024-08-04
>
> Dear reviewer XhGj, we thank you for your valuable comments and review. We are glad to hear that you have enjoyed reading our paper. Below, we provide our response to your comments and questions.
>
> - **W1: Shifted source distribution during inference.**
> As you have pointed out, the shifted source distribution might affect the performance of our method. However, as studied in depth in Appendix A and B of our paper, our method is still guaranteed to work under a mild assumption, and under general non-deterministic preferences. In particular, our method is theoretically guaranteed to work if the support of the distribution $p_{0}$ (of less preferred samples) is included in the support of the distribution $p_{1}$ (of more preferred samples). This means that as long as the preference is continuous, we can guarantee that for any samples $y \sim \pi_{\mathrm{ref}}$, there is a non-zero probability of being included in the negatively labeled dataset. We have also provided empirical evidence for our choice of source distribution $\pi_{\mathrm{ref}}$ instead of $p_{0}$ in Appendix B of our paper. We will clarify this assumption in the main paper, as well as include further details in the limitation section of our paper.
> - **W2: Experiments for the NLP domain.**
> We have obtained a promising result in one of the popular NLP tasks, one that is also adopted in the DPO paper. Please refer to the above common response section. We believe that the results above will serve as strong evidence and support that our method can be applied to general language-based tasks.
> - **Q1: Inference on D4RL tasks.**
> For the inference on the D4RL tasks, we sample a reference trajectory $\tau$ using a copy of the current environment, with the same initial state. In scenarios where using a copy of such an environment is available (e.g., simulations), this is no problem, except for increased computation costs. However, as you pointed out, in some real-world applications, we may need an additional dynamics model in order to compute a reference trajectory. We will include this as one of our limitations. We would like to note here that this limitation is only for the case of RL tasks, and not for the image and language domains.
> - **Q1-1: Inference on D4RL tasks.**
> Once the reference trajectory $\tau$ is obtained, we then apply PFM to obtain an improved trajectory $\tau^{+}$. In our implementation, we choose only the first action from the improved trajectory during inference. Of course, one may choose all actions from the improved trajectory to reduce the computational cost incurred during flow-matching, at the expense of performance loss. Hence, there is a trade-off between the computation (for flow-matching) and the performance gain, and one may find an appropriate intermediate as a design choice, suitable for their task.

---

> > ### Comment · Reviewer_XhGj · 2024-08-08
> > **Thanks for your response!**
> >
> > Thanks for your response and hard work. After reading your rebuttal and other reviews, I've decided to maintain my score.

---

> > > ### Author Response · Authors · 2024-08-14
> > >
> > > We thank you again for your time and effort. We will incorporate your comments and feedback in the revision.

---

### Official Review · Reviewer_yRMJ · 2024-07-13

**Soundness:** 4
**Presentation:** 4
**Contribution:** 4
**Rating:** 9
**Confidence:** 4

**Summary:**

The paper presents a new framework for preference-based reinforcement learning, relying on Flow Matching. This framework utilizes flow-based models with optimal transport that interpolate between less preferred data and more preferred data, eliminating the need to estimate a reward function implicitly or explicitly. The authors provide a rigorous framework supported both by formal proofs and clear experimental evidence of the method's performance and demonstrate the effectiveness of iterative PFM, which iteratively narrows down the distribution towards preferred outputs.

**Strengths:**

The paper notably excels along each of the dimensions:
- originality: the paper presents a novel approach to preference alignment that hasn't been explored in the literature and shows a very promising potential for further exploration.
- quality: the problem is well positioned, claims in the paper are clearly stated and the flow of argument circles back to support the claims throughout the paper. The authors do a good job of illustrating the nature of RLHF overfitting by starting with deterministic preferences, which are easy to illustrate and extend to non-deterministic cases. Then they prove that the proposed objective (12) for the marginal distribution $p1$ obtained from the preference model $\mathbf{P}(y>y')$ is robust to overfitting even in the deterministic case.
- clarity: the flow of argument is delivered with exceptional clarity. Notably, the authors provide both formal proofs of theorems and intuitive explanations, e.g. in lines 204-205 " marginalization iteratively "narrows" down the distribution towards the outputs with higher preference", which are then supported by the results in Table1, demonstrating performance improvements with lower variance across all baseline methods and supporting the claim in lines 36-37 "By simply adding a preference flow matching module to black-box models, 37 PFM eliminates the need for fine-tuning the black-box model itself, providing a significant advantage."
- significance: the contributions of the paper are significant yet the impact on more complex tasks is yet to be explored. In the current form, the method is not applicable to NLP tasks as LLMs, but flow matching per se is capable of working on multiple modalities and textual data.




Demonstrate

**Weaknesses:**

The paper is very strong, no notable weaknesses observed, suggestions for improvements are provided in the next section.

**Questions:**

- Consider an expansion of failure mode explanations for DPO from the paper https://arxiv.org/pdf/2404.10719, namely susceptibility to distribution shift, which can cause issues with generalizability in the regions without data support. They tie back to the author's critique of the failure modes of DPO, like overfitting.

- line 210: "Is PFM beneficial than methods optimizing for explicit/implicit reward model" is this missing "more beneficial"? (same for Q3)

- lines 126-129 highlight that the starting distribution $p_0$ is inaccessible during the inference step, and instead, they simply start from $y \sim \pi_{ref}$ as the starting point. This is an important implication for sampling/inference. Thus, a comment explaining why this works and whether it can be applied in any scenario under consideration is crucial. Consier is adding this to limitations if there are issues with applicability.

- In the conclusion and limitations, the authors state limitations of applicability to NLP tasks: " Future research should explore ways to adapt the PFM 314 framework for variable-length data, potentially through innovative alignment techniques or alternative 315 frameworks suited for text generation tasks." Diffusion Forcing: https://arxiv.org/abs/2407.01392 has been introduced recently to combine both autoregressive next token predictions with full-sequence diffusion. Consider this as a minor suggestion given the time constraints and timelines of the initial submission, the could be a useful source for future work.

**Limitations:**

The authors adequately addressed the limitations of their work throughout the paper and in the conclusions and limitations section.

A minor suggestion:
- The authors state the challenges of existing preference-based RL methods such as scalability, inefficiency, and the need for model modifications, especially with black-box, and later address them in the paper, except for scalability, which calls for stating more clear limitations or suggesting lines of future work to address this (extending the limitations in lines 309-310 would suffice).

---

> ### Author Rebuttal · Authors · 2024-08-04
>
> Dear reviewer yRMJ, we appreciate your invaluable comments and feedback. Below, we provide our response to your comments and questions.
>
> - **Q1. Shifted source distribution during inference.**
> As mentioned in our paper, since the source distribution $p_{0}$ is inaccessible during inference, we instead start from a shifted source distribution $\pi_{\mathrm{ref}}$. This can indeed be a factor that may deteriorate the performance of our method. However, as discussed thoroughly in Appendix A and B of our paper, it is still theoretically guaranteed to work if the support of the distribution $p_{0}$ (of less preferred samples) is included in the support of the distribution $p_{1}$ (of more preferred samples). In other words, for any sample $y \sim p_{1}$, as long as it has a non-zero probability of being labeled as less preferred (i.e. $p_{0}(y) > 0$), we can guarantee that for any samples $y \sim \pi_{\mathrm{ref}}$, there is a non-zero probability of being included in the negatively labeled dataset. Hence, we can guarantee a non-zero probability of existing learned flow from that sample $y \sim \pi_{\mathrm{ref}}$ to a more preferred sample. That being said, a failure regime of our method may include a scenario where the distribution of the ground-truth measure of the preference scores is sparse, that is, only the extremes (either very preferred or nearly non-preferred) are present in the dataset. We will include content in the limitation section.
> - **Q2. Future works for NLP domains.**
> Thank you for pointing out a useful source (Diffusion Forcing) for future work. We also believe that these lines of work can be adapted to our framework and applied to NLP tasks. Aligned with this direction, we have shown a promising result of our method on an NLP task; please refer to the common response above.

---

### Official Review · Reviewer_vT4A · 2024-07-23

**Soundness:** 3
**Presentation:** 2
**Contribution:** 2
**Rating:** 5
**Confidence:** 5

**Summary:**

This paper introduces Preference Flow Matching (PFM), a novel framework for preference-based reinforcement learning that addresses key limitations of existing methods like Reinforcement Learning from Human Feedback (RLHF). PFM leverages flow matching techniques to directly learn from preference data. The core innovation of PFM lies in its formulation of preference alignment as a problem of learning a flow between marginal distributions of less preferred and more preferred samples, moving away from the implicit discriminative classification formulation of DPO into a discriminative/generative formulation. Methodologically, PFM employs a conditional flow matching objective to learn the vector field representing the preference flow. It also does not require the Bradley-Terry assumption. PFM is also compatible with 'reflow' like the original flow matching formulation. The authors derive the marginal distribution and prove it converges to a maximizing uniform. This all represents a clear, if straightforward, combination of (reward-free) preference learning and flow-matching providing a ready-made comparison with existing *-PO methods.

**Strengths:**

The paper has multiple strengths. The combination of preference learning with flow matching is clear, if straightforward ; and its exposition is sound and easy to follow. The authors provide solid theoretical analysis, including proofs of optimality and convergence. This gives the method a strong mathematical grounding, which is often lacking in more heuristic approaches. In theory, PFM tackles important issues in existing methods, particularly the problem of reward overfitting and the challenges of aligning black-box models.

**Weaknesses:**

- Paper writing could be improved as several typos remain ('A careful reader might noticed' l.126, 'ordinary diffusional (sic) equation' l.89...). Some proofreading is in order !

- Most importantly, the empirical contribution is not overly compelling. It is surprising that on MNIST the DPO results are that poor, both in absolute terms and relatively (worse than RLHF). IPO is not considered, DPO is probably taken offline rather than online (which generally tends to improve results), and most importantly, the beta parameter is not iterated upon/optimized (if it is, then this needs to be made clear, if possible in the main text, but it is not even in Appendix D.1). It is well known that the beta parameter is crucial for the performance of those methods and one advantage of very small scale experiments such as MNIST is that iterating is inexpensive. Tuning the baseline to the maximum would lend credence to these experiments; besides, it is done in the other suite of experiments.

- Similarly, the offline RL experiments are small scale (Gym-MuJoCo. This time the beta parameter is optimized, and on all expert domains, the PFM performance is not massively different from DPO - in fact, the expert average for DPO fine-tuned is stronger than that of PFM (even though PFM provides variance reduction). While we know that DPO has tendencies to overfit and reward-hack, this is noted by the authors themselves who ask in section 5.3 'is learning a flow truly beneficial' ? Finding this detailed and balanced discussion in the paper was a nice display of intellectual fortitude and very welcome. Nonetheless, the empirical benefits of the proposed approach so far are quite ambiguous. The paper would truly benefit from showcasing one indisputable and conclusive empirical setting in favour of PFM.

- Finally, the paper doesn't deeply explore the computational costs of PFM compared to other methods, which could be an important practical consideration. In particular, is reflow worth it ?

**Questions:**

- What stopped the authors (other than lack of time, infrastructure, compute) from trying for small-scale language modelling experiments, since this is the domain that originally stemmed the derivation of DPO and IPO ? In reference to 'Another notable limitation is its applicability to the natural language processing (NLP) domain. The PFM framework, as currently designed, cannot be directly applied to NLP tasks because the source and target texts may have different lengths', it would technically be possible to model distributions over fixed length sequences, along with padding and/or an EOS token. One great advantage of working in the text domain is the ability to display sample completions from finetuned models in a qualitative study. Also, testing with actual human preferences would actually be better.

- Even if additional experiments are out of scope of this current review period, which other empirical designs can the authors think of that would better showcase the benefits of PFM compared to *-PO ?

**Limitations:**

Limitations are noted in the 'weaknesses' section below and mostly acknowledged in the paper.

---

> ### Author Rebuttal · Authors · 2024-08-04
>
> Dear reviewer, we appreciate your thorough review and valuable feedback. Below are our responses to your comments and questions.
>
> - **W1: Weak empirical contribution.**
> We have obtained promising results in a new domain, please refer to the above common response section for our results in the selected NLP task. We believe this new result is strong enough to prove the empirical usefulness of our framework.
> - **W2: Regarding experimental results for D4RL tasks.**
> As you pointed out, PFM may not show significant improvement over DPO in D4RL benchmarks. However, it is important to note that we achieved similar performance increases from the reference policy without any fine-tuning of the original policy. Hence, the greatest advantage of our method is its ability to be attached to any black-box model for improved preference alignment. Combined with the new empirical results obtained in an NLP task, and experiments in the vision domain with MNIST examples, we believe it provides a comprehensive advantage for our framework in general preference alignment.
> - **W3: Regarding MNIST results.**
> The RLHF and DPO results of MNIST experiments are obtained from the best working $\beta$ found from the same search space as in the D4RL tasks. We thank you for pointing out this missing detail; we will add this to our paper.
> - **W4: Regarding computational cost.**
> For the computational cost incurred during the training phase, PFM is significantly more efficient than existing frameworks, as it only requires training a small add-on module. For example, please refer to the above common response section to see the difference in required parameter sizes for training in even a simple NLP task. Due to the limited time, we were not able to directly compare the training cost of RLHF fine-tuning and PFM, since we did not fine-tune the reference language models on our own, but adopted an open-source fine-tuned model that is already publically available. During inference, there is negligible computational complexity due to the small size of the attached PFM module. We believe this additional computational cost during inference will be even more negligible as the size of the base reference model increases.
>
> For your questions regarding the NLP domains and additional experiments, we believe that our new results provided in the common response will address your concerns and demonstrate the applicability of our proposed methods across various scenarios. Thank you once again for your valuable feedback.

---

> ### Comment · Reviewer_vT4A · 2024-08-14
>
> Thank you for your rebuttal and additional empirical results. Cognizant of the fact these take time to run, we have to stress the scale-dependent nature of those results and would reserve judgment for now as to whether the billion parameters+ LLM scale will improve as much as GPT-2. This leaves the empirical section fairly ambiguous in our view, and as such, we maintain our score.

---

### Author Rebuttal · Authors · 2024-08-04

# (Common Response) New Domain: Added NLP Task

We applied our method (PFM) to the new NLP domain. We adopt a controlled (positive) sentiment review generation task, which is one of the main tasks previously tackled by the DPO paper. As done in the DPO paper, to perform a controlled evaluation, we adopt a pre-trained sentiment classifier as the preference annotator from huggingface library. The preference dataset is constructed from randomly selected pairs of moview reviews $y^{+}, y^{-}$ from the well-known IMDB dataset, where the preference is obtained from the classifier logit probability $p(\mathrm{positive} | y^{+}) > p(\mathrm{positive} | y^{-})$. We then train our PFM model on this preference dataset, to obtain the marginal distribution of the preferred (positive sentiment) review $p_{1}(y^{+})$.

For our PFM framework to be applied to variable-length inputs, we employ a T5-based autoencoder from a huggingface library to obtain fixed-sized (1024-dimensional vector) embedding of the input texts, allowing us to work within the fixed-size latent space. Once the embeddings $z^{+}$ and $z^{-}$ are obtained for each text sample $y^{+}$ and $y^{-}$, we learn the conditional flow using PFM from $z^{-}$ to $z^{+}$. During inference, we apply PFM to the latent embedding $z$ of the given input text $y$, and decode the improved latent embedding using the T5 decoder. We adopt the same U-Net architecture used in our MNIST experiment, where we reshape the 1024-dimensional vector into a two-dimensional (32, 32) image tensor.

Note that the traditional approach, including standard RLHF frameworks and DPO, involves fine-tuning the LLMs on this preference dataset. Due to the LLMs' large number of parameters, fine-tuning these models is generally a computationally expensive task. However, we emphasize here that our PFM method requires a relatively smaller network than the original reference policy (LLM). We summarize below the number of parameters that require training for each framework in tackling this task. As we will discuss later on, PFM requires training a much smaller number of parameters (around 1.2%) while still achieving better performance. Furthermore, PFM does not require fine-tuning the original reference model, (and keeps the original model’s representation) as PFM can be attached to any existing black-box model as an add-on module.

| GPT-2 (RLHF/DPO fine-tuning) | U-Net (PFM) |
| --- | --- |
| 124M | 1.5M |

For our pre-trained reference policy for inference, we adopt the GPT-2 SFT model on the IMDB dataset from the huggingface library. We also compare our method with the RLHF fine-tuned model from the same huggingface source. Below, we report the average preference score (from the classifier annotator) of 100 randomly generated test samples (reviews) for each method. Please refer to the attached PDF files for detailed visualization of the score distributions. As shown in the table below, PFM is able to improve the preference score of any baseline model to which it is attached. Notably, iterative PFM with only five iterations achieves the best performance over all the baselines. We also note here that the PFM is trained with the original dataset, not by the dataset generated from the RLHF fine-tuned policy. In other words, PFM can be trained on the base dataset generated from the SFT model, and still be attached to arbitrary fine-tuned policies to further improve its performance. Due to the limited time constraints, we were not able to obtain results for DPO fine-tuned policy, as we could not find a DPO fine-tuned policy that is publically available. However, we believe that our results will also be comparable with DPO.

| Reference | PFM | Iterative PFM (5 iter) |
| --- | --- | --- |
| -0.3607 | 0.6399 | **2.7469** |
| **RLHF fine-tuned** | **RLHF + PFM** | **RLHF + Iterative PFM (5 iter)** |
| 2.0156 | 2.178 | **2.7894** |

We also compute the win rate with GPT-4 evaluation. Please refer to the attached PDF for the prompts we used to generate win rate. Both PFM and RLHF (PPO) fine-tuned policy excels the reference (SFT) policy with win rates 100%. We also compare the win rates of our various PFM models with the RLHF fine-tuned policy. Notably, we observe that the iterative PFM with 5 iterations on SFT model outperforms the RLHF fine-tuned policy. If PFM is added on top of the RLHF fine-tuned policy, we observe near 100% win rates for both RLHF + PFM and RLHF + Iterative PFM. See the below table for the summarized results. Note that this win rate is also correlated with the preference scores provided by the pre-trained classifier.

|  | SFT + PFM | SFT + Iterative PFM (5 iter) | RLHF + PFM | RLHF + Iterative PFM (5 iter) |
| --- | --- | --- | --- | --- |
| RLHF (PPO) | 2% | 85% | 99% | **100%** |

Interestingly, we observe that the distribution of the scores tends to shift more toward that of the preferred samples with an increasing number of PFM iterations. (Please refer to the histogram provided in the attached PDF.) This result aligns with our theoretical insights: the PFM framework learns to shift the source distribution (i.e., the sampled distribution of the reference policy) toward the marginal distribution of the more preferred samples.

If time permits, we plan to further experiment on additional NLP tasks before the camera-ready deadline. Currently, being in the initial stages, we have only demonstrated the add-on capability of PFM with GPT-2. As future work, we aim to apply our method to larger models like GPT-4 and tackle more complex tasks. Recent studies have been actively exploring language generation using diffusion models in continuous latent spaces, such as latent diffusion for language generation. Additionally, as reviewer yRMJ suggested, there are promising studies that combine diffusion and autoregressive generation. By integrating our method with these approaches, we anticipate broader applicability and increased usage of PFM in the language domain.

---

### Decision · Program_Chairs · 2024-09-25

**Decision:**

Accept (poster)

**Comment:**

This papers proposes Preference Flow Matching (PFM), a new approach for preference-based RL that doesn't require an explicit (or implicit) reward model, leverating optimal transport to shift the original sampling distribution towards that of preferred samples.

Reviewers' ratings exhibited a high variance (3 to 9), with a common main concern regarding the lack of application to NLP tasks (which are the main reason behind the current high interest in preference-based learning). However, the authors provided new experimental results on an NLP task that at least partially alleviate this concern.

One reviewer (2tji / rating 4) still remained not entirely convinced after extensive discussions with the authors, in particular regarding the relevance of the full set of experimental results. I do agree with the reviewer that there is room for improvement in this area, however, I also want to show appreciation for the originality of this work that offers a fresh approach on preference-based learning, which I believe may serve as source of inspiration to future developments in similar directions. Considering that all reviewers were appreciative of this novel contribution, which is clearly presented and motivated, I consider this paper worth accepting at NeurIPS.

(I also note that the lowest-scoring reviewer -- with a 3 -- did not acknolwedge the authors' response, in spite of my reminder to do so)